# JMJD6 participates in the maintenance of ribosomal DNA integrity in response to DNA damage

**Jérémie Fages**[1], **Catherine Chailleux**[1], **Jonathan Humbert**[2], **Suk-Min Jang**[2], **Jérémy Loehr**[3], **Jean-Philippe Lambert**[3], **Jacques Côté**[2], **Didier Trouche**[1ʘ], **Yvan Canitrot**[1ʘ] *

**1** LBCMCP, Centre de Biologie Intégrative (CBI), Université de Toulouse, CNRS, UPS, Toulouse, France, **2** Centre de Recherche sur le Cancer de l'Université Laval, axe Oncologie du Centre de recherche du CHU de Québec-Université Laval, Québec, Canada, **3** Centre de Recherche sur le Cancer de l'Université Laval, axe Endocrinologie et néphrologie du Centre de recherche du CHU de Québec-Université Laval, Québec, Canada

ʘ These authors contributed equally to this work.
* yvan.canitrot@univ-tlse3.fr

**Data Availability Statement:** All MS files used in this study were deposited at MassIVE (http://massive.ucsd.edu) and at ProteomeXchange (http://www.proteomexchange.org/). They were

## Abstract

Ribosomal DNA (rDNA) is the most transcribed genomic region and contains hundreds of tandem repeats. Maintaining these rDNA repeats as well as the level of rDNA transcription is essential for cellular homeostasis. DNA damages generated in rDNA need to be efficiently and accurately repaired and rDNA repeats instability has been reported in cancer, aging and neurological diseases. Here, we describe that the histone demethylase JMJD6 is rapidly recruited to nucleolar DNA damage and is crucial for the relocalisation of rDNA in nucleolar caps. Yet, JMJD6 is dispensable for rDNA transcription inhibition. Mass spectrometry analysis revealed that JMJD6 interacts with the nucleolar protein Treacle and modulates its interaction with NBS1. Moreover, cells deficient for JMJD6 show increased sensitivity to nucleolar DNA damage as well as loss and rearrangements of rDNA repeats upon irradiation. Altogether our data reveal that rDNA transcription inhibition is uncoupled from rDNA relocalisation into nucleolar caps and that JMJD6 is required for rDNA stability through its role in nucleolar caps formation.

## Author summary

Ribosomal DNA is composed of repeated sequences and is the most transcribed genomic region. Transcription of rDNA is essential for cellular homeostasis and cell proliferation. Numerous pathologies such as cancer and neurological disorders are associated with defective rDNA repeats maintenance. The mechanisms involved in the control of rDNA integrity involve major DNA repair pathways such as Non-Homologous End Joining and Homologous Recombination. However, how they are controlled and orchestrated is poorly understood. Here, we identified JMJD6 as a new member of the maintenance of

assigned the identifiers MassIVE MSV000083409
and PXD012603.

**Funding:** This work was supported by grants from
the Fondation ARC to DT (programme ARC), from
the Ligue National Contre le Cancer to DT (Equipe
labellisée), from Canceropole GSO and EDF
(comité Radioprotection) to YC, from the Canadian
Institutes of Health Research (FDN-143314) to JC
and from the Natural Sciences and Engineering
Research Council of Canada (Discovery grant
RGPIN-2017-06124) and a John R. Evans Leaders
fund from the Canada Foundation for innovation
(37454) to J-PL. J-PL holds a Junior 1 salary
award from the Fonds de Recherche du Québec-
Santé (FRQ-S) and JC the Canada Research Chair
on Chromatin Biology and Molecular Epigenetics.
The funders had no role in study design, data
collection and analysis, decision to publish, or
preparation of the manuscript.

**Competing interests:** The authors have declared
that no competing interests exist

rDNA integrity. We observed that JMJD6 controls the recruitment of NBS1 in the nucleolus in order to lead to the proper management of rDNA damages

## Introduction

Cells are continuously exposed to DNA damages which are repaired by different DNA repair pathways according to the nature of the damage (double- or single-strand breaks, base modifications,. . .). DNA double-strand breaks (DSBs) are among the most deleterious DNA damage and cells have developed two main pathways to repair them: homologous recombination (HR) and non-homologous end joining (NHEJ)[1]. Moreover, the deleterious effects of DNA damage also depend on where damages occur within the genome with damages in transcribed regions being potentially the most detrimental. In this study, we focused on ribosomal DNA (rDNA) the most transcribed genomic locus. It is composed of tandem repeats (200–300) present on the five acrocentric chromosomes in human cells [2]. In interphase cells, rDNA localises within specialized subcompartments, the nucleoli, in which rDNA transcription and rRNA processing take place.

Maintaining the number of rDNA repeats as well as their level of transcription is crucial for cellular homeostasis [3]. Instability of the rDNA repeats was reported in cancer [4] and Bloom and ataxia-telangectasia cell lines [5,6]. Therefore, DSBs generated in rDNA need to be efficiently and accurately repaired to prevent inaccurate recombination events that could result in loss of repeats in such repetitive regions. To date, repair of DSBs occurring at rDNA repeats is poorly understood and the mechanisms involved remain controversial [7,8]. Indeed, some studies have proposed that rDNA DSB repair is mediated by NHEJ within the nucleolus [9,10]. However, persistent breaks are relocalised to nucleolar caps at the periphery of the nucleolus. This process is ATM-dependent and is tightly linked to transcription inhibition [9,10,11,12]. Those breaks directed to nucleolar caps may possibly be more accessible to nucleoplasmic repair proteins and may be repaired by the action of HR proteins shown to be present at the nucleolar periphery throughout the cell cycle [13]. Lastly, a distinct nucleolar DNA damage response involving ATR activation and an ATM-TCOF1-MRN axis was identified to mediate rDNA transcription inhibition concomitantly to nucleolar restructuring in response to rDNA induced DSB [12,14] and under osmotic stress [15].

DNA repair is influenced by the chromatin context that can be specified by histone post translational modifications [16,17,18]. In this study, we identify JMJD6, a member of the histone demethylase family, as an important player of the response to DNA damage occurring in rDNA. JMJD6 is a member of the Jumonji C domain-containing proteins family and has been described as presenting two enzymatic activities, an arginine demethylase contributing to histone methylation control [19] and a lysyl hydroxylase activity targeting non histone proteins such as p53 [20] and the splicing factor U2AF65 [21]. Recently, an additional function for JMJD6 as a tyrosine kinase targeting the Y39 tyrosine of the histone variant H2A.X has been reported in triple negative breast cancer cell lines overexpressing JMJD6 [22]. JMJD6 was also shown to modulate the DNA damage response independently of its enzymatic activity through the modulation of histone H4 acetylation. The authors report that its depletion leads to increased DNA double strand breaks repair and resistance to ionizing radiations [23]. Here we show that JMJD6 is recruited to DNA damages generated in the nucleolus and is important for their repair, therefore favouring rDNA arrays stability. Furthermore we show that JMJD6 is necessary for the relocalisation of the repair factor NBS1 in the nucleolus upon DNA damage, thus providing novel insights into the mechanisms underlying repair of rDNA DSBs.

## Results

### JMJD6 is involved in the DNA damage response to ionizing radiation

In order to obtain an integrated view of the involvement of histone demethylases in the response to DNA damage, we performed a screen using a siRNA library directed against all known or putative (Jumonji domain-containing proteins) histone demethylases in U2OS cells. Using as a read-out the DNA damage γH2AX foci formation, we identified JMJD6 as a hit whose depletion caused higher γH2AX staining following ionizing radiation (IR) exposure. To confirm these observations and to rule out possible off-target effects, we used two additional siRNAs directed against JMJD6, which efficiently decreased JMJD6 expression (Fig 1C). We observed that transfection of any of these siRNAs increased γH2AX staining both before and in response to IR (Fig 1A and 1B). Taken together, these data indicate that JMJD6 is required for the normal response to irradiation-induced DNA damage in U2OS cells. We then assayed the effect of JMJD6 depletion on the two major DSB repair pathways, Non-Homologous End Joining (NHEJ) and Homologous Recombination using cell lines with specific reporter systems. JMJD6 depleted cells did not exhibit defects on HR efficiency (Fig 2A). However, NHEJ

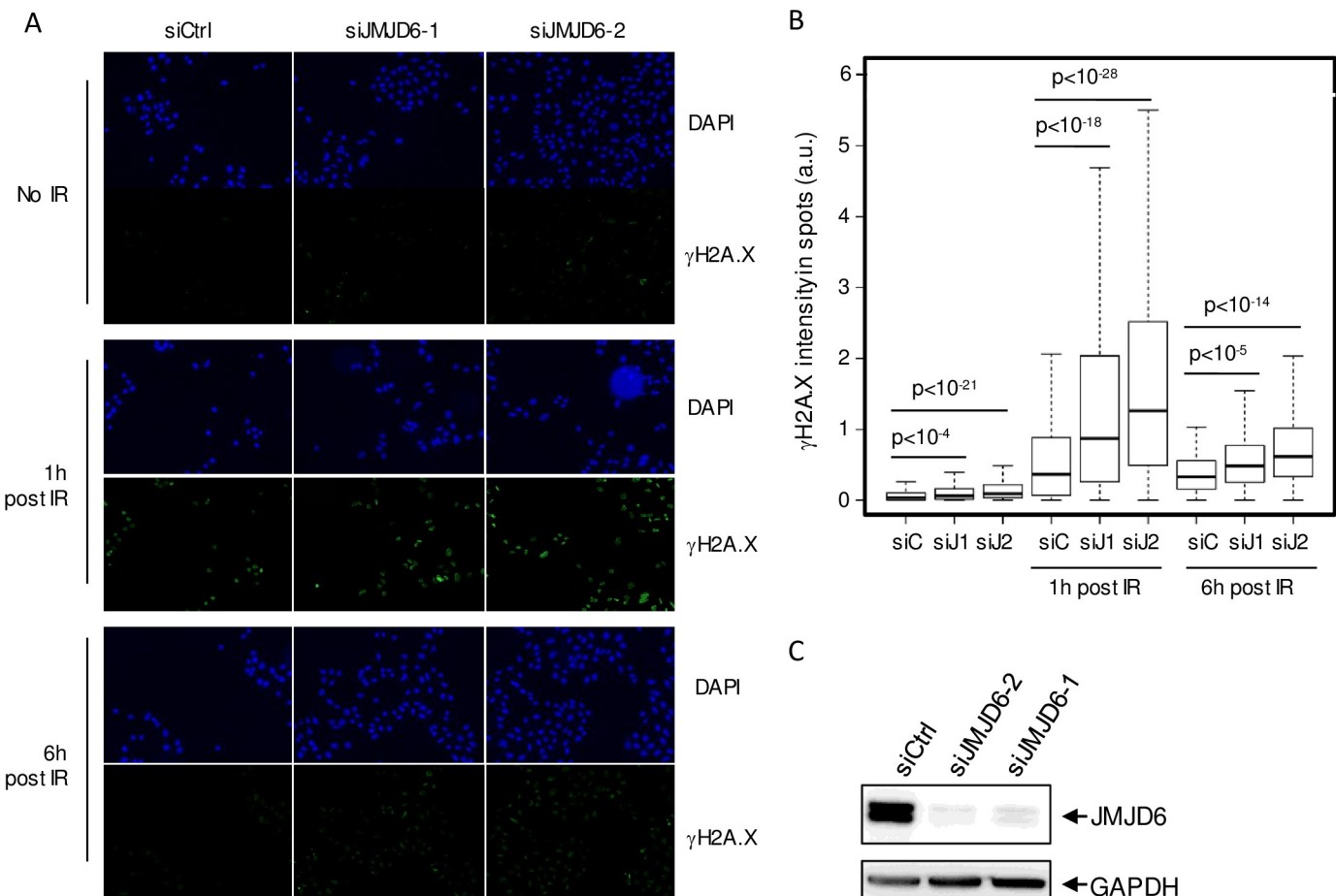

**Fig 1. JMJD6 expression is required for the normal DNA damage response to ionizing radiation exposure.** (A). Images of U2OS cells transfected with the indicated siRNA, exposed to ionizing radiations (8 Gy) and subjected to DAPI and gH2AX staining one hour or six hours following irradiation, as indicated. Scale bar 50 μm. (B). Quantification of gH2AX staining by high throughput microscopy in cells from A. A minimum of 200 cells were quantified for each conditions. A representative experiment from 3 is shown. The p values of the difference between the indicated samples are shown (Wilcoxon test). (C). siRNA efficacy tested through Western blot analysis on whole cell extracts.

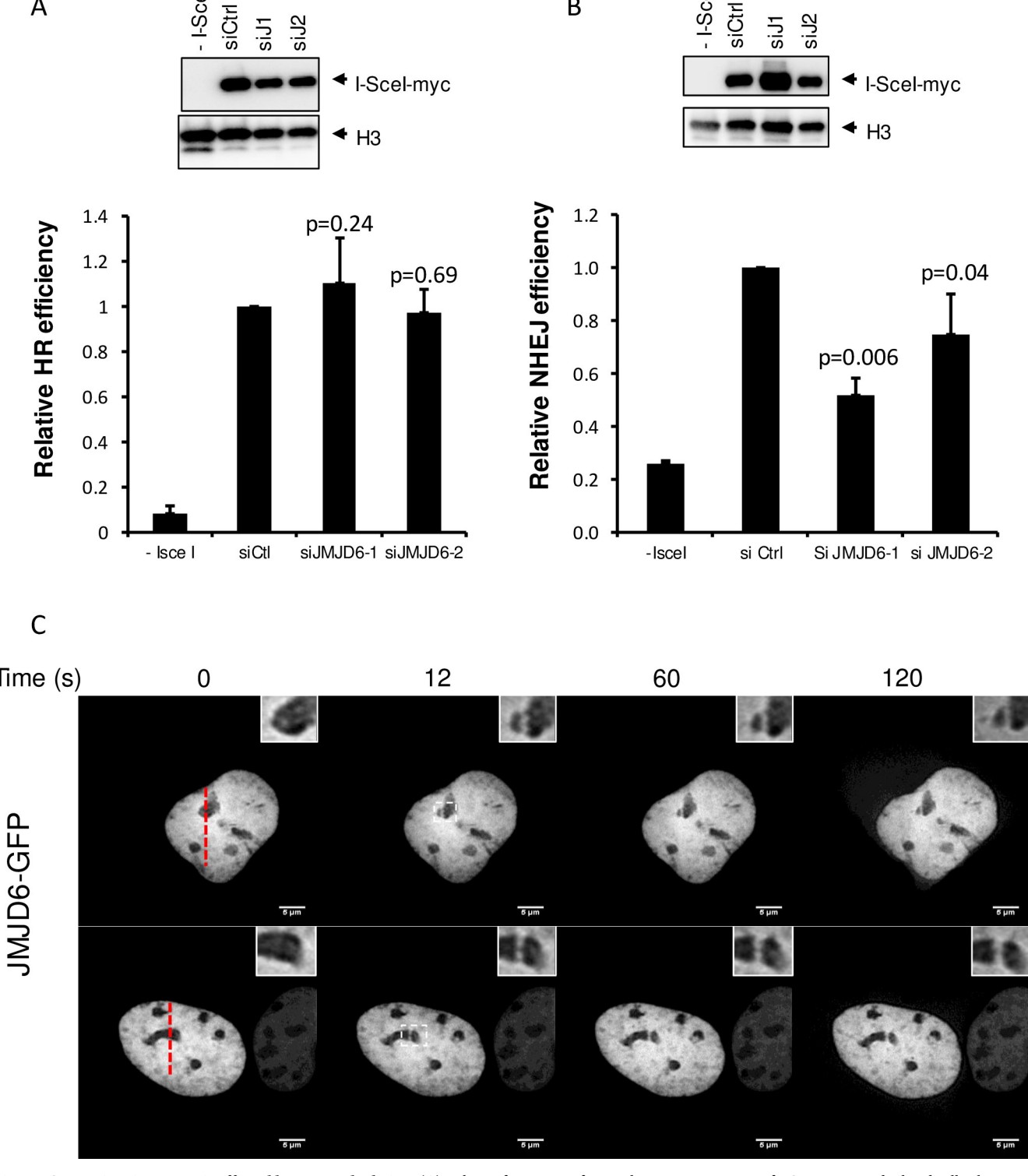

**Fig 2. DSB repair using NHEJ is affected by JMJD6 depletion.** (A) Relative frequency of Homology-Driven Repair of DSB in JMJD6 depleted cells. the percentage of GFP positive cells was measured by flow cytometry and calculated relative to 1 in cells transfected by the control siRNA. The mean and standard deviation from four entirely independent experiments are shown. The right panel shows a Western blot monitoring I-SceI expression in the different transfected cells. (B) Same as in A except that GCS5 human fibroblasts containing a reporter system for NHEJ were used. The p values of the difference to the control siRNA are indicated (Student t test). (C) U2OS cells transfected by a plasmid expressing JMJD6-GFP fusion protein and subjected to local laser irradiation 24h later. Images of cells before and after the indicated time following nucleus laser irradiation are shown. The red dashed lines represent the laser irradiated region in the nucleus. insert: magnification of the nucleolus irradiated regions. Scale bar 5 μm.

efficiency was significantly decreased (Fig 2B), indicating that JMJD6 expression is required for NHEJ.

We next tested whether the effect of JMJD6 on DNA damage signaling and repair could be direct by assessing the recruitment of GFP-tagged JMJD6 to sites of laser induced-DNA damage using live cell imaging. We consistently observed a strong enrichment of JMJD6-GFP at DNA damage induced in the nucleolus (Fig 2C) which was not detected when using 53BP1-GFP, as a positive control, or GFP alone as a negative control (S1A and S1B Fig). In addition, the recruitment of JMJD6-GFP at DNA breaks in the nucleolus was quicker (around 1-2min) than that of 53BP1 in the nucleoplasm (10 min). Of note, in some cells expressing GFP-tagged JMJD6, we found some enrichment of JMJD6 at nucleoplasm sites of DNA damage (S1C Fig). Note that this was not observed in all cells whereas JMJD6-GFP was systematically recruited to damage induced in the nucleoli. JMJD6-GFP recruitment to these latter damages was also observed in MRC5 cells, a non transformed cell line (S1D Fig). Thus, these data indicate that JMJD6 is rapidly recruited to DNA damage sites which occur in the nucleolus, strongly suggesting that it could play a direct role in DNA damage signaling and repair following breaks occurring at nucleolar DNA. We also investigated the recruitment of JMJD6 to endonuclease-mediated DSBs by ChIP experiments. We constantly observed the presence of JMJD6 at rDNA in the absence of breaks induction. However, we did not observe any further enrichment in the vicinity of induced DSBs (S2 Fig). This result could be due to a rapid and transient recruitment of JMJD6 to DSBs at the rDNA. Because of the repetitive nature of rDNA, the presence of JMJD6 at uncut repeats could mask its recruitment to breaks induced at only few copies. Nevertheless, these data show a recruitment of JMJD6 at rDNA, consistent with a role in rDNA damage management.

## Cells deficient for JMJD6 are sensitive to DNA damage occurring within rDNA

We next tested whether JMJD6-depleted cells, which have an impaired DNA damage response, are sensitive to irradiation by performing clonogenic cell survival assays post-irradiation. We generated JMJD6 KO cell lines (JMJD6 -/-) by CRISPR-mediated genome editing in U2OS cells. We found that these cells are more sensitive to IR than parental cells (Fig 3A, see S3A and S3B Fig for another clone of KO cells). Importantly, these JMJD6 KO cells complemented with the expression of JMJD6 from a plasmid were less sensitive than KO cells, regaining nearly equivalent sensitivity to IR than parental cells (Fig 3A and S3C and S3D Fig). Thus, these data indicate that the defect in the DNA damage response and repair observed upon JMJD6 knock down using siRNA translates into increased sensitivity to DNA damages in JMJD6-negative cells. We also re-introduced in JMJD6 KO cells a mutant, which harbors 3 point mutations in the catalytic domain of JMJD6 preventing Fe(II) binding [19]. This mutant disrupts both the hydroxylation and demethylase activity, given that hydroxylation is the first step towards demethylation. Interestingly, cells complemented with this mutant form of JMJD6 (KO+Mut) showed no complementation for sensitivity to IR (Fig 3A) although it was recruited at laser-induced damage as observed for the WT form of JMJD6 (S4 Fig). JMJD6 was also shown to harbor tyrosine kinase activity [22]. However, in this study, no mutant having lost this activity while retaining the hydroxylase/demethylase activity is described [22], precluding the investigation of the specific role of its kinase activity.

The recruitment of JMJD6 at nucleolar damage sites suggests that it could be involved in rDNA damage response and repair. To test whether JMJD6 depletion could lead to increased sensitivity to damages generated specifically in the nucleolus, we carried out a cell survival assay following the targeted induction of rDNA breaks using CRISPR-cas9 and a guide RNA

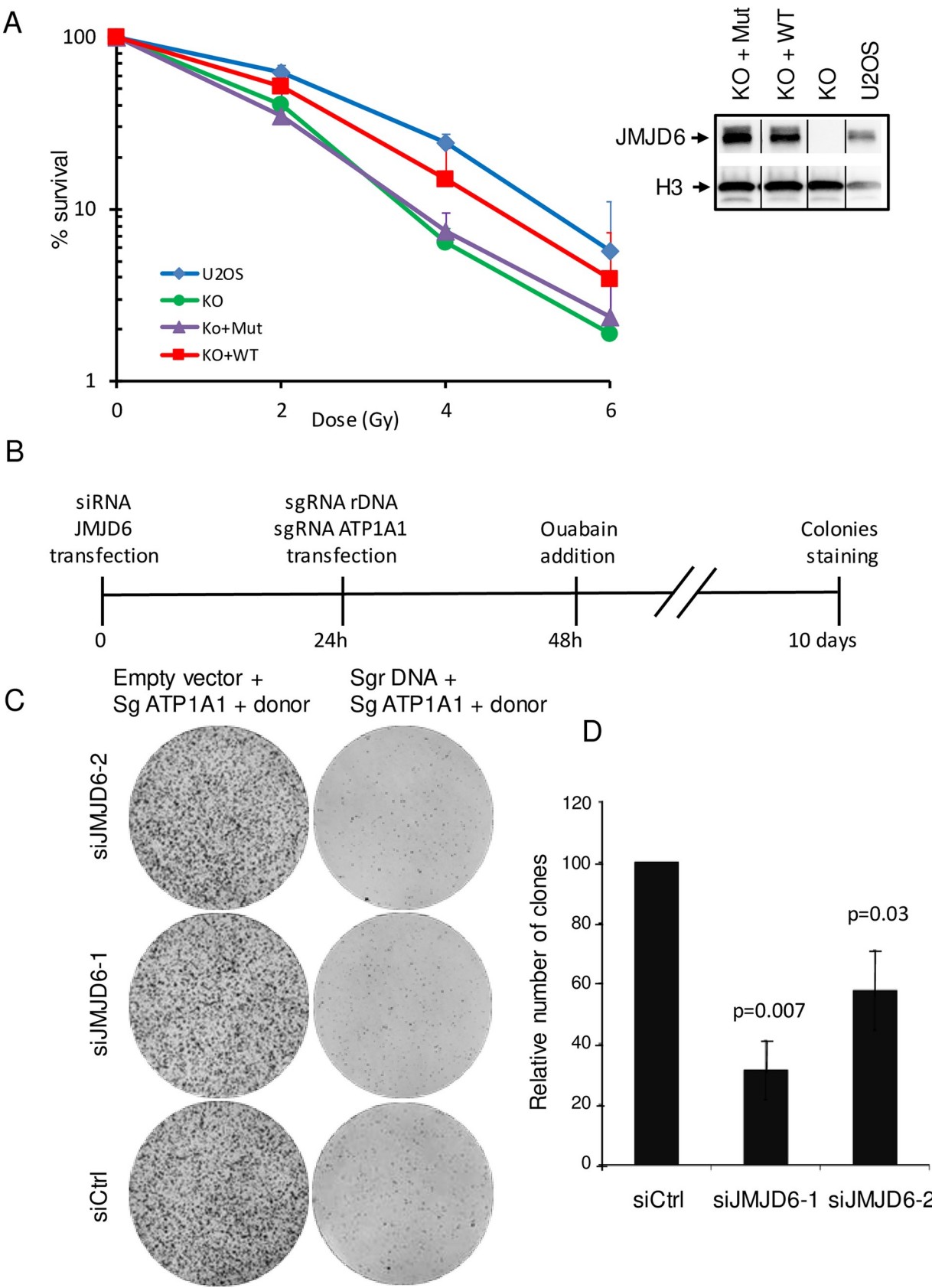

**Fig 3. JMJD6-depleted cells are more sensitive to DSBs induced in rDNA.** (A) Clonogenic cell survival assay performed on U2OS cells, JMJD6-KO and JMJD6-KO- complemented cell lines with WT (KO+WT) and catalytic inactive forms of JMJD6 (KO+Mut) and exposed to increased doses of irradiation. The mean and standard deviation from three independent experiments are plotted. A similar experiment with another clone is shown in S3 Fig. Expression of JMJD6 in the different cell lines examined by Western blot. The bar indicates that the original image was cut to remove unnecessary lanes. (B) Scheme of the CRISPR experiment designed to test the sensitivity of JMJD6 depleted cells subjected to targeted rDNA breaks. U2OS cells were transfected with a vector expressing a guide RNA targeting the rDNA (sgrDNA) together with a guide RNA targeting the ATP1A1 gene (sgRNA ATP1A1) and a donor DNA rendering ATP1A1 resistant to ouabain. As a control an empty vector not coding for guide RNA targeting rDNA but with a guide RNA targeting the ATP1A1 gene (sgRNA ATP1A1) and a donor DNA rendering ATP1A1 resistant to ouabain was used. 10 days later, clones were stained with crystal violet. (C) Example of a typical experiment. (D) Quantification of the experiment, the number of clones was calculated relative to 100, with 100 corresponding to cells transfected by the control siRNA. The mean and standard deviation from three independent experiments are plotted. The p values of the difference between the siJMJD6 samples and the Ctrl siRNA sample are indicated (Student t test).

targeting rDNA [13]. In this experiment, we included a control guide RNA targeting the ATP1A1 gene coding for the Na/K pump together with a donor DNA, which renders genome-edited cells resistant to ouabain [24] (Fig 3B). By performing clonogenic experiments in the presence of ouabain, we could thus rule out any effect of JMJD6 depletion on Cas9 or guide RNA expression. In the absence of control guide RNA targeting the ATP1A1, ouabain treatment killed all cells, whereas transfection of this guide led to the generation of many ouabain resistant clones, reflecting targeted genome editing at the ATP1A1 gene (Fig 3C, left). Co-transfecting the guide RNA targeting rDNA strongly diminished the number of ouabain-resistant colonies (Fig 3C, right), indicating that DSBs in rDNA impaired cell survival. Interestingly, depletion of JMJD6 further decreased cell survival upon CRISPR-induced rDNA breaks (Fig 3C and 3D). Taken together, these data indicate that the full expression of JMJD6 is specifically required for the management of DSBs occurring at the rDNA locus, consistent with our finding that JMJD6 can be recruited to the nucleolus upon DNA damage.

## JMJD6 depletion generates genetic instability at rDNA repeats

Faithful repair of DSBs occurring in rDNA allows the maintenance of rDNA repeats integrity. rDNA repeats are present on the five acrocentric human chromosomes and can be visualized in metaphase as ten UBF foci, which correspond to the Nucleolus Organizer Regions (NORs) (Fig 4A) [9,13]. To study the involvement of JMJD6 in preserving rDNA stability JMJD6 KO or control cells were subjected or not to IR (2Gy) followed by a 30h recovery period before scoring the number of NORs using UBF as a marker (Fig 4A and 4B). As expected, the median number of NORs in control cells was around ten. In JMJD6 KO cells, a significant decrease was already observed in absence of DNA damage (median number of 9) meaning that JMJD6 is necessary even in absence of external DNA damage. After IR exposure an additional loss of NORs was observed (Fig 4B). This decrease in NOR number corresponds to *a bona fide* loss of rDNA sequences since UBF expression was unchanged (Fig 4C). Moreover, it does not result in an off-target modification of the KO cell line since the number of NORs was largely preserved in JMJD6 complemented-KO cells (Fig 4B). Thus, these data indicate that JMJD6 expression is required for the maintenance of the number of NORs upon irradiation, indicating its major role in the maintenance of rDNA repeats integrity. To better characterize the genetic instability at rDNA in response to DNA damage we performed DNA FISH combing using fluorescent FISH probes targeting rDNA (Fig 4D). Example of normal rDNA repeats organized in head-to-tail (regular succession of red-green units) is shown and ascribed as canonical. Example of rDNA rearrangements identified as non-canonical are pointed by a star. Results show that in absence of external DNA damage JMJD6 depletion *per se* induced a higher level of rDNA rearrangements (Fig 4E). In response to induced-DSB we observed an increase in rDNA rearrangements in control cells which was higher in JMJD6-depleted cells.

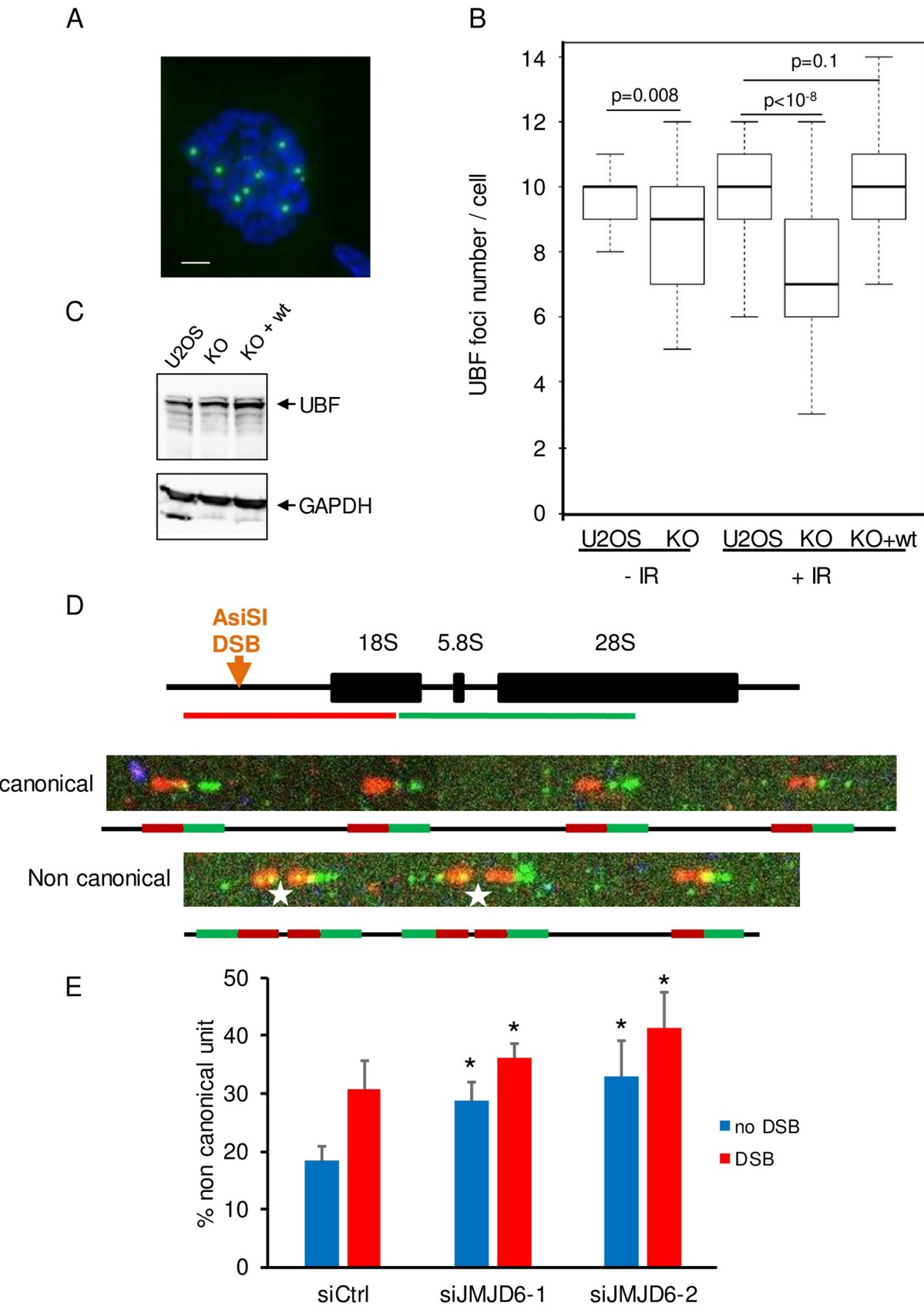

**Fig 4. JMJD6 expression is required for rDNA repeat integrity following DNA damage** (A) Representative image showing individual NORs in a U2OS cell in metaphase stained using an anti UBF antibody. Scale bar 5 μm. (B) Ionizing radiation (2 Gy) exposure of U2OS cells, U2OS cells inactivated for JMJD6 expression (KO) and a clone from the latter cell line in which wild type JMJD6 was reintroduced (KO + wt). The number of UBF foci in cells was then counted and the results represented as box plot. For each point a minimum of 50 metaphases were scored. Results from one representative experiment from 2 independent experiments is shown. The p values of the difference between the indicated samples are shown (Wilcoxon test). (C) Western blot analysis of UBF expression in the different cell lines. (D) Evaluation of rDNA rearrangements by FISH combing. Representation of a rDNA repeat with the position of the DSB induced by AsiSI after OHTam treatment. The green and red lines represent the FISH probes used in DNA FISH combing experiments and targeting two adjacent sequences in the rDNA. An example of a canonical array (without rDNA rearrangement) is shown. Note that the green and red probes are in the same order throughout the array. An example of a non-canonical (with rDNA rearrangement indicated by a star) rDNA repeat is also shown. E. Quantification of non-canonical rearrangements measured before and after DSB induction in siRNA control and siRNA JMJD6-depleted cells. Quantification was performed on duplicate samples with more than 400 units examined on each samples. Results are the mean +/- s.e.m. of three independent experiments. * p<0.1 was considered as significant. p values of the difference in non induced DSB were calculated using Student t test and are p = 0.04 and p = 0.08 for siJMJD6-1 and siJMJD6-2, respectively. p values of the difference after DSB are p = 0.076 and p = 0.073 for siJMJD6-1 and siJMJD6-2, respectively.

Together these results confirm that JMJD6 is important to preserve rDNA from major rearrangements.

## JMJD6 interacts with Treacle and controls the NBS1-Treacle interaction in response to DNA damage

We next investigated the mechanism by which JMJD6 could influence DNA repair in the nucleolus. To characterize the JMJD6 interactome, we raised a K562-derived cell line in which the endogenous JMJD6 gene was edited with CRISPR/cas9 to produce a C-terminal 3xFlag/2xstreptavidin-tagged JMJD6 protein. After tandem affinity purification from nuclear soluble extracts, we examined JMJD6-interacting proteins by mass spectrometry. Consistent with the known role of JMJD6 in mRNA splicing [25,26], we recovered many proteins linked with the spliceosome apparatus and mRNA 3'-end processing (CPSF, SYMPK, DDX41, WDR33), validating our experimental strategy (S1 Table). In addition, our results showed proteins specific for the nucleolar compartment and associated with rDNA transcription (Fig 5A). Among them, TCOF1 (Treacle) drew our attention as it participates in the response to DNA damage in the nucleolus [27,28]. Importantly, this interaction was also observed by BioID when using TCOF1 as a bait, indicating that the two proteins are very close in the cells (Fig 5B and S2 Table). We confirmed the JMJD6-TCOF1 interaction in U2OS cells by performing Proximity Ligation assay (PLA) (Fig 5C and 5D) and co-immunoprecipitation (Fig 5E) and found with confocal microscopy that they co-localise in the nucleolus (S5 Fig).

Next we checked the effect of JMJD6 depletion on TCOF1 function. Immunofluorescence analysis, showed that TCOF1 expression and localisation were not significantly altered in JMJD6 KO cells (S6B and S6C Fig). Interestingly, TCOF1 has been reported to interact with NBS1, allowing its relocalisation into the nucleolus in response to DNA damage to repress rDNA transcription [27,28]. We thus performed a PLA assay for monitoring Treacle-NBS1 interaction after DNA damage according to JMJD6 status. Using the JMJD6 KO cell line and the WT or Mutant -complemented cell lines, we observed a higher level of interaction between NBS1 and Treacle in JMJD6 KO and Mutant-complemented cell lines compared to WT-complemented cell line after IR exposure (Fig 6 and S7 Fig). The proximity of the three proteins was confirmed by confocal microscopy in NBS1-GFP transfected cells (S8 Fig). These results suggest that JMJD6 controls the extent of the NBS1-Treacle interaction.

## JMJD6 depletion does not relieve rDNA transcription repression

The TCOF/NBS1 complex has been shown to mediate transcriptional silencing of rDNA repeats upon DNA breaks induced at rDNA. To analyse the effect of JMJD6 depletion on

A

**Without DNA damage**

| Total peptides | Symbol |
|---|---|
| 50 | JMJD6 |
| 32 | CPSF1 |
| 32 | SYMPK |
| 25 | DDX41 |
| 16 | WDR33 |
| 15 | TCOF1 |

**After DNA damage**

| Total peptides | Symbol |
|---|---|
| 62 | JMJD6 |
| 35 | CPSF1 |
| 32 | SYMPK |
| 31 | DDX41 |
| 25 | WDR33 |
| 11 | TCOF1 |

B

**Without DNA damage**

| Total peptides | Symbol |
|---|---|
| 403 | POLR1A |
| 176 | POLR1B |
| 78 | TOPBP1 |
| 26 | JMJD6 |

**After DNA damage**

| Total peptides | Symbol |
|---|---|
| 444 | POLR1A |
| 200 | POLR1B |
| 95 | TOPBP1 |
| 22 | JMJD6 |

C

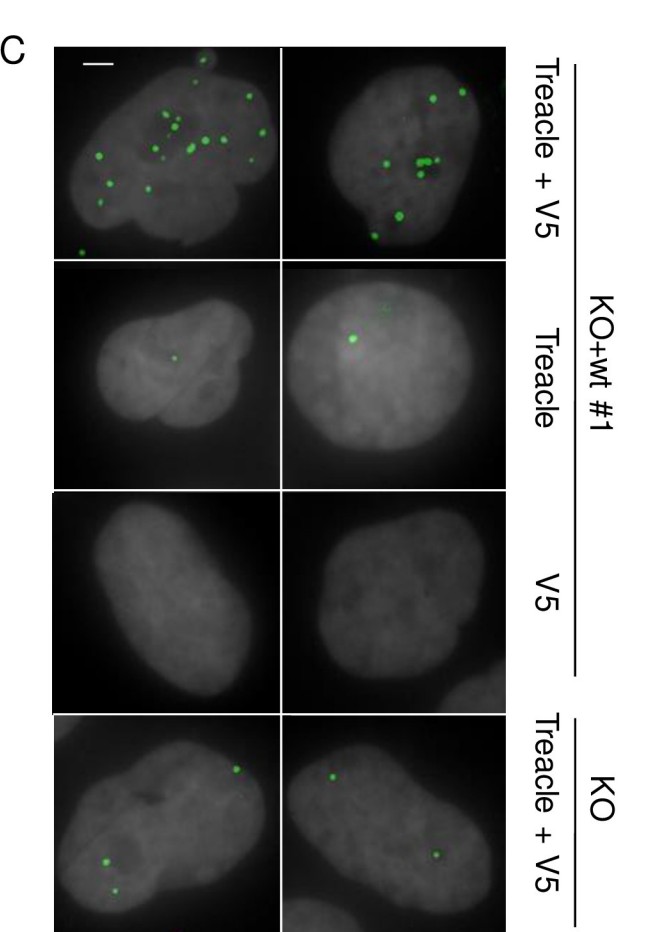

D

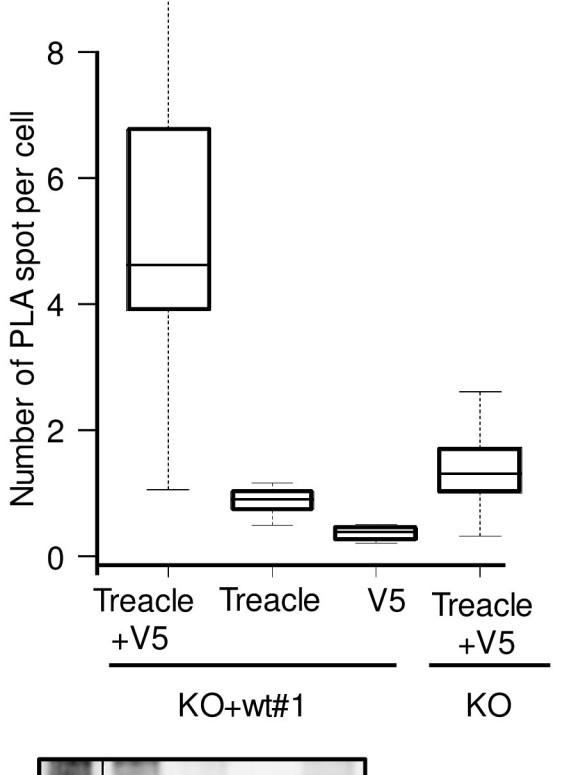

E

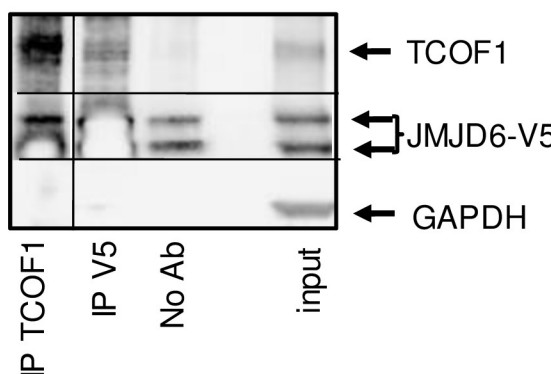

**Fig 5. JMJD6 interacts with TCOF1 (Treacle).** (A) List of JMJD6 interactors identified after mass spectrometry (total spectral counts) after tandem affinity purification of endogenous JMJD6 isolated from K562 cells before and after DNA damage induced with etoposide (20μM 1h). See S1 Table for complete results. (B) Selected list of interaction partners associated with TCOF1 detected by BioID with and without etoposide treatment. See S2 Table for complete results. (C) Images of JMJD6-V5/Treacle (TCOF1) interaction revealed by Proximity Ligation Assay (PLA) in cells exposed to IR (5 Gy, 1h post IR). Scale bar 5 μm. (D). Quantification of PLA-induced spot for the different conditions of antibodies was performed on at least 200 cells for each conditions. (E) Co immunoprecipitation of TCOF1 with JMJD6. Total cell extracts from U2OS cells were immunoprecipitated with TCOF1 or V5 antiboby or no antibody as a control and revealed by Western blot for Treacle and JMJD6-V5 presence. The bars indicate that the original image was cut to remove unnecessary lanes. Note that the ECL substrate was exhausted in the V5 Western blot in the V5 and TCOF1 IP because of the very strong signal given by V5-tagged JMJD6 in the V5 IP or the immunoglobulin in the TCOF1 IP.

rDNA transcription we used the DIvA cell line in which DSBs are produced across the genome by the AsiSI endonuclease [29] one of which being located within rDNA and potentially generating one DSB per rDNA repeat [30]. As expected, control cells showed a decrease in rDNA transcription following DSB induction, as measured by the incorporation of 5-FUrd metabolic labelling (Fig 7A and 7B). Interestingly, rDNA transcription was further decreased in JMJD6-depleted cells compared to control cells (Fig 7) while rDNA transcription in the absence of DNA damage remained largely unaffected (although in some experiments such as the one shown in Fig 7C, we could observe a slight decrease of basal rDNA transcription upon JMJD6 knock-down). Similar effects on rDNA transcription repression were observed 1h post IR (8 Gy), with a further reduction of transcription in JMJD6 depleted cells (Fig 7C and 7D). Interestingly, rDNA transcription levels had fully recovered 6 hours following IR in both control and JMJD6-depleted cells (Fig 7C and 7D), indicating that the rDNA transcription decrease observed upon JMJD6 depletion was transient. We confirmed these results in the MRC5 cell line in which we generated specific DNA damage in rDNA using CRISPR-Cas9 (S9 Fig). Altogether, these data show that JMJD6 expression defect leads to a slightly more efficient transcriptional repression upon DNA breaks induction. This is consistent with the increase in TCOF1/NBS1 interaction we observed in the absence of JMJD6, given that this complex mediates rDNA transcriptional repression. Note however that we cannot rule out the possibility that this higher transcriptional repression is due to higher levels of unrepaired DNA damage in the absence of JMJD6.

## JMJD6 is required for nucleolar caps formation

The Treacle/NBS1 complex was also shown to be important for the relocalisation of rDNA at the nucleolar periphery in structures called nucleolar caps, upon induction of DSBs in rDNA [9,13]. We thus tested the involvement of JMJD6 in this process by detecting nucleolar caps by UBF staining [30]. As expected, upon DSB induction in rDNA using DIvA cells, nucleolar caps were readily formed in a significant proportion of cells (Fig 8A). Strikingly, less cells displaying nucleolar caps were observed in JMJD6 depleted cells compared to control cells (Fig 8B). This decrease in nucleolar caps formation was not caused by an altered expression of UBF (Fig 8C), nor by changes in cell cycle distribution (S10 Fig). Similar results were obtained after exposure to a high dose of IR (20 Gy 6h) showing that the defective generation of nucleolar caps in JMJD6 depleted cells was not restricted to endonuclease-generated DNA damages (S11 Fig). More importantly, these results were confirmed by using the CRISPR-Cas9 system inducing targeted DSB at rDNA (Fig 8D and 8E). This result indicates that it is not due to a signaling from DNA breaks induced outside the nucleoli. JMJD6 depletion by siRNA also affected nucleolar caps produced upon induction of breaks in the rDNA in the immortalized MRC5 cell line, demonstrating that it is not restricted to U2OS cells (Fig 8E). Altogether, these data indicate that JMJD6 expression is required for the formation of nucleolar caps. Importantly, nucleolar caps have been proposed to be associated with rDNA transcription inhibition in response to DNA damage [11]. However, here we show that nucleolar caps formation and

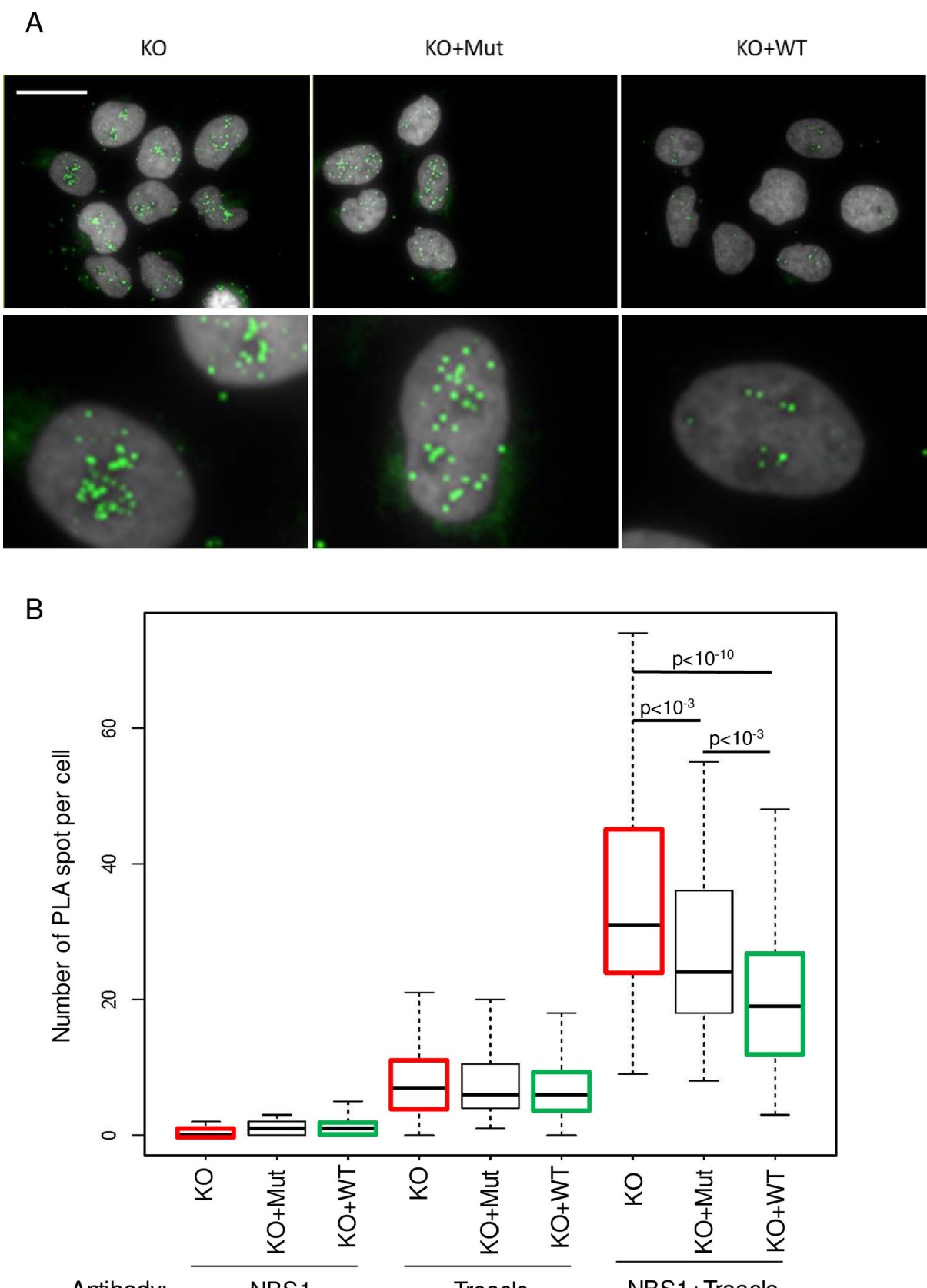

**Fig 6. JMJD6 controls the interaction between Treacle and NBS1.** (A) Images of NBS1/Treacle interaction revealed by Proximity Ligation Assay (PLA) in cells exposed to IR (5 Gy, 1 h post IR). Scale bar 50 μm. The bottom panels show magnifications from selected regions from these images.(B) Quantification of PLA-induced spot for the different conditions of antibodies was performed on at least 200 cells for each conditions. One representative experiment from four independent experiments is shown. Two other experiments are shown in S7 Fig. The p value of the difference between KO and KO+Mut to KO+WT are shown.

transcription inhibition can in fact be uncoupled since JMJD6 depleted cells display transcriptional repression, with yet less nucleolar caps when measured in identical conditions (in DiVA cell lines treated 4 hours with OHTam to induce AsiSI DSB).

Previous studies showed the importance of ATM signaling for the generation of nucleolar caps and transcription inhibition [9,13]. We thus investigated the relationship between JMJD6 and ATM in the pathway leading to nucleolar caps. First, we tested the activation of ATM in the different cell lines in response to IR. As shown in the Fig 9A, ATM was normally activated in WT- and JMJD6-KO cell lines in response to IR. Then, we tested whether ATM signaling and JMJD6 were in the same pathway for the generation of nucleolar caps. As previously described for JMJD6 knockdown cells (Fig 8B), JMJD6 KO cells were defective for nucleolar caps formation. JMJD6-KO cells complemented with the WT form of JMJD6 presented more nucleolar caps, this increase being sensitive to ATM inhibition (Fig 9B). Thus, these data indicate that JMJD6 and ATM are in the same pathway. Considering that ATM activation is normal upon JMJD6 inactivation and that the ATM-dependent transcription inhibition process is not abolished upon JMJD6 knock-down, these data suggest that JMJD6 lies downstream of ATM in the pathway leading to nucleolar caps formation.

## JMJD6 affects the recruitment of NBS1 into the nucleolus in response to DNA damages

DNA damages in the nucleolus need to be repaired before restart of the transcription. To repair DSB cells can use HR or NHEJ. The most rapid pathway to repair DSB is the use of NHEJ but for complex lesions or DSB requiring faithful repair cells can use homologous recombination. Both processes are influenced by NBS1 via its involvement in the MRN complex. Because we observed that JMJD6 interact with TCOF1 we wondered whether JMJD6 could alter NBS1 recruitment into the nucleolus.

To test whether JMJD6 exerts a similar role to TCOF1 in the control of NBS1 localisation, we assayed the presence of NBS1-GFP foci in the nucleolus of JMJD6 KO cells in response to IR (Fig 10A). As expected, we observed an increase in NBS1 foci in the nucleolus of control cells in response to IR exposure. In contrast, IR-induced nucleolar localization of NBS1 was strongly decreased in JMJD6 KO cells, an effect which was reversed when these cells were complemented with JMJD6 but not in the cells complemented with the demethylase-defective form of JMJD6 (Fig 10B). These results are not the consequence of altered expression or cellular localization of TCOF1 in the JMJD6 KO cell line (S6B and S6C Fig). In addition, we observed that NBS1 and TCOF1 co-localize in JMJD6 KO cells (S6D Fig). These results show that JMJD6 and its enzymatic activity are required for the recruitment of NBS1 to the nucleolus in response to DNA damage, providing a direct link between JMJD6 and the presence of DNA repair factors in the nucleolus.

## Discussion

Here, we demonstrate for the first time that the JMJD6 histone demethylase is important for the response to DSBs occurring at rDNA repeats in the nucleolus. This conclusion relies on four independent lines of evidence: first, JMJD6 is mobilized upon DNA damages induced in

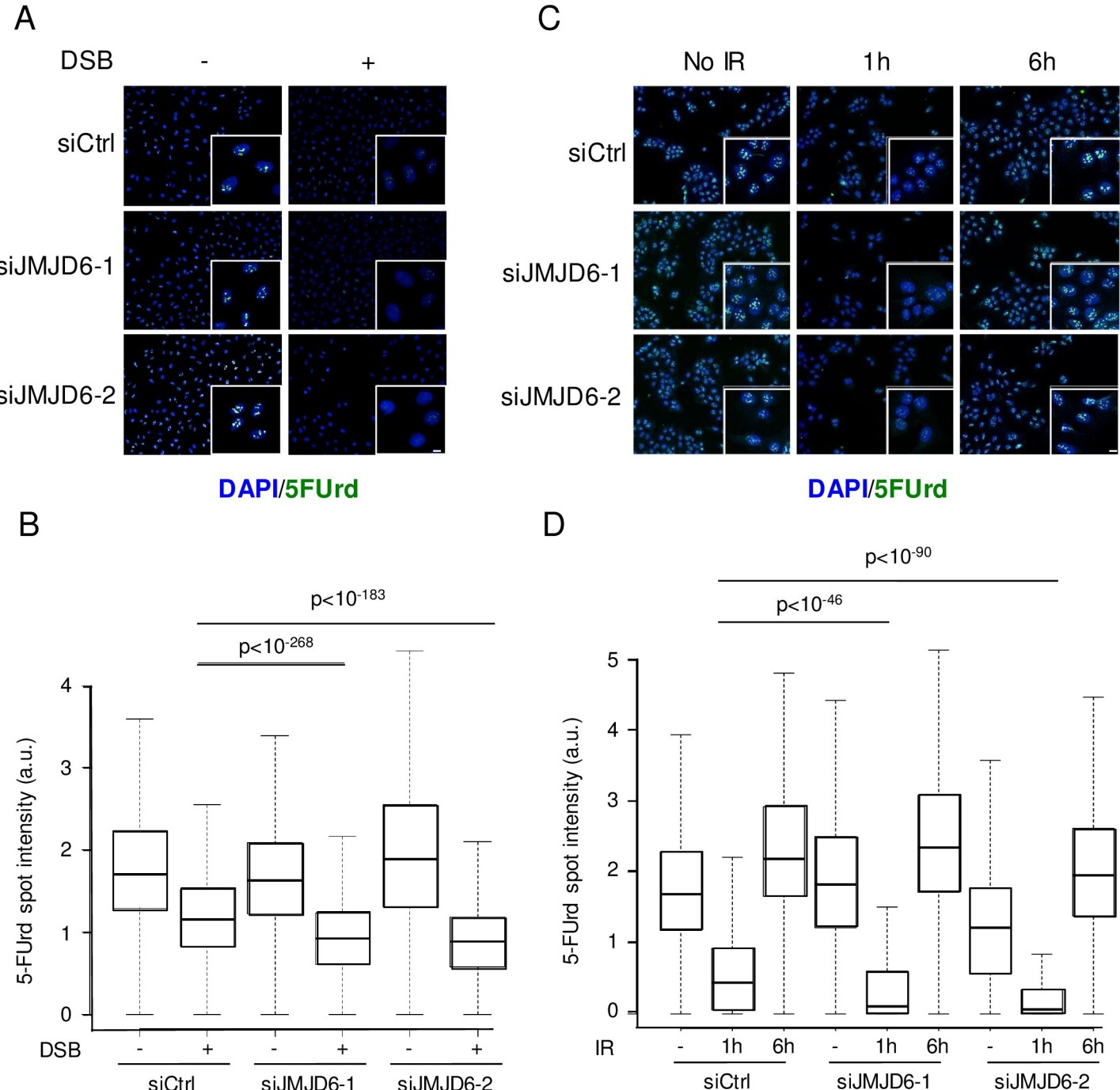

**Fig 7. JMJD6 expression favours rDNA transcription following DNA damage.** (A) Level of rDNA transcription assessed through 5FUrd incorporation following 4 hours of endonuclease-mediated DSBs (AsiSI) in U2OS DIVA cells transfected with the indicated siRNA, insert: a magnified region of the image. Scale bar 5 μm. (B) Quantification of 5FUrd staining by high throughput microscopy in cells from A. A representative experiment from 2 independent experiments is shown. The p values of the difference between the indicated samples are shown (Wilcoxon test). (C) Same as in A, except that cells were exposed to ionizing radiation (IR) (8 Gy) or not and stained 1 hour or 6 hours following irradiation. Scale bar 5 μm. (D) Quantification of 5FUrd staining by high throughput microscopy in cells from C. A representative experiment from 2 independent experiments is shown. For each data point a minimum of 200 cells were quantified. The p values of the difference between the indicated samples are indicated (Wilcoxon test).

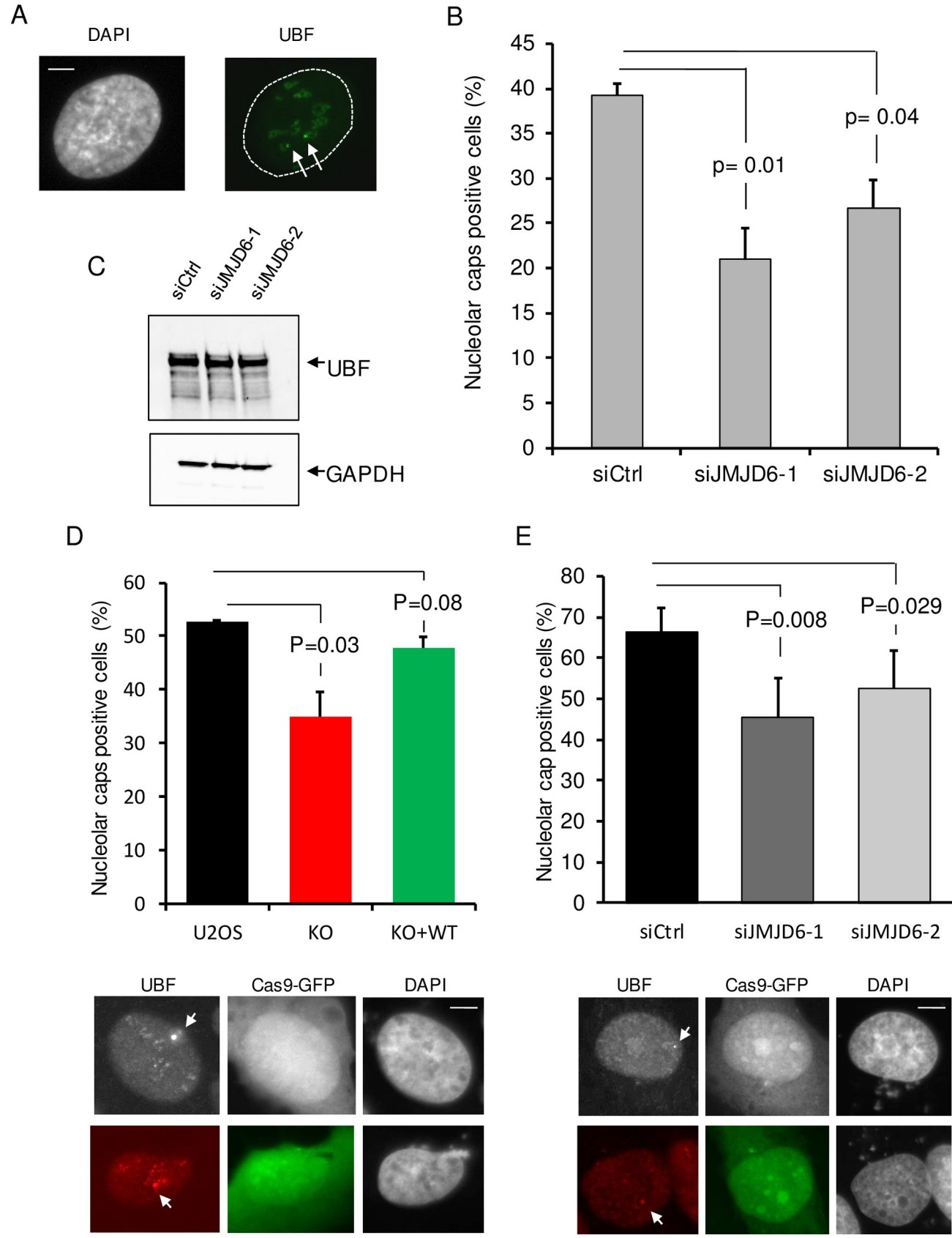

**Fig 8. Nucleolar caps formation in response to DNA damage.** (A) Images from endonuclease-induced DSB (4 hours of OHTam treatment) in DIvA cells showing nucleolar caps revealed by UBF foci formation at the nucleolus periphery. The arrows point towards nucleolar caps. Scale bar 5 μm. (B) Quantification of nucleolar caps positive cells after JMJD6 depletion using siRNA. (C) Western blot analysis of UBF expression after siRNA-induced JMJD6 depletion. (D) Quantification of nucleolar caps positive cells in parental U2OS and JMJD6 KO and complemented (KO+WT) cell lines after generation of rDNA DSB using CRISPR-Cas9. Images of nucleolar caps revealed by UBF staining in Cas9-GFP transfected cells. The arrows point towards nucleolar caps. (E) Quantification of nucleolar caps positive cells after JMJD6 depletion using siRNA in the MRC5 cell line and generation of rDNA DSB using CRISPR-Cas9. Images of nucleolar caps revealed by UBF staining in Cas9-GFP transfected cells. The arrows point towards nucleolar caps. The mean and standard deviation from three independent experiments are shown. For each conditions a minimum of 100 cells were counted. The p values of the difference between the indicated samples are indicated (Student t test).

the nucleolus. Second, JMJD6-depleted cells are more sensitive to DSBs specifically occurring in rDNA. Third, JMJD6 depleted cells harbour genetic instability at rDNA loci upon DNA break induction. Fourth, JMJD6-depleted cells are defective for the formation of nucleolar caps in which repair of persistent DSBs by homologous recombination is generally thought to occur.

Importantly, JMJD6 influences transcription [31]. We thus cannot rule out the hypothesis that some of the functions of JMJD6 in rDNA breaks management are due to indirect effect on the expression of repair proteins. We tested the mRNA levels of several repair proteins and none of them were significantly affected in JMJD6 KO cells (S6A Fig). In addition, the expression and localization of two essential components of the cellular response to nucleolus DNA damage, NBS1 and Treacle [12,14,28] are not altered (S6B and S6C Fig). Moreover, two lines of evidence point towards a direct role of JMJD6 in rDNA breaks management: Firstly, JMJD6 is recruited to DNA breaks induced in the nucleolus. Secondly, JMJD6 physically interacts with TCOF1, a nucleolar protein whose function in rDNA breaks management is proven [12,14,28].

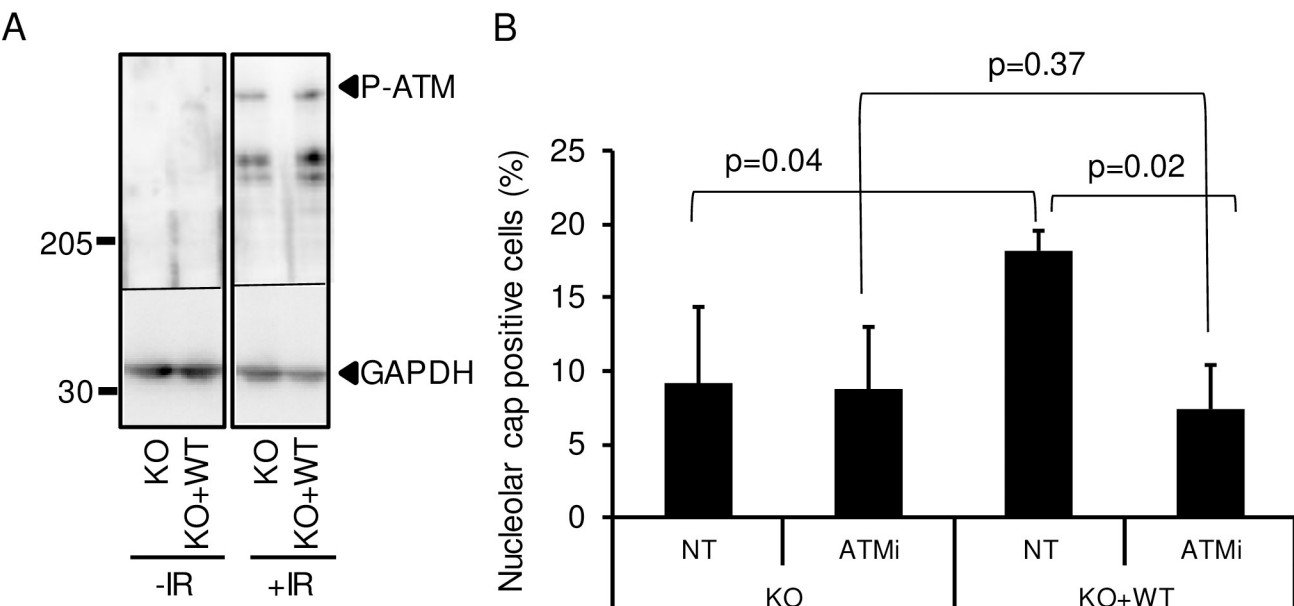

**Fig 9. ATM activation in JMJD6 defective cells.** (A) Western blot analysis of ATM activation in response to IR (20 Gy 1h post IR) performed by the detection of the phosphorylated form of ATM. The smaller bands induced upon irradiation are probably shorter forms of P-ATM. (B) Dependence on ATM activity for the formation of nucleolar caps in response to IR. Cells were treated or not with the ATM inhibitor. A minimum of 100 cells were counted for each condition in each independent experiment. Results are the mean +/- standard deviation from three independent experiments. The p values of the difference between the indicated samples are indicated (Student t test).

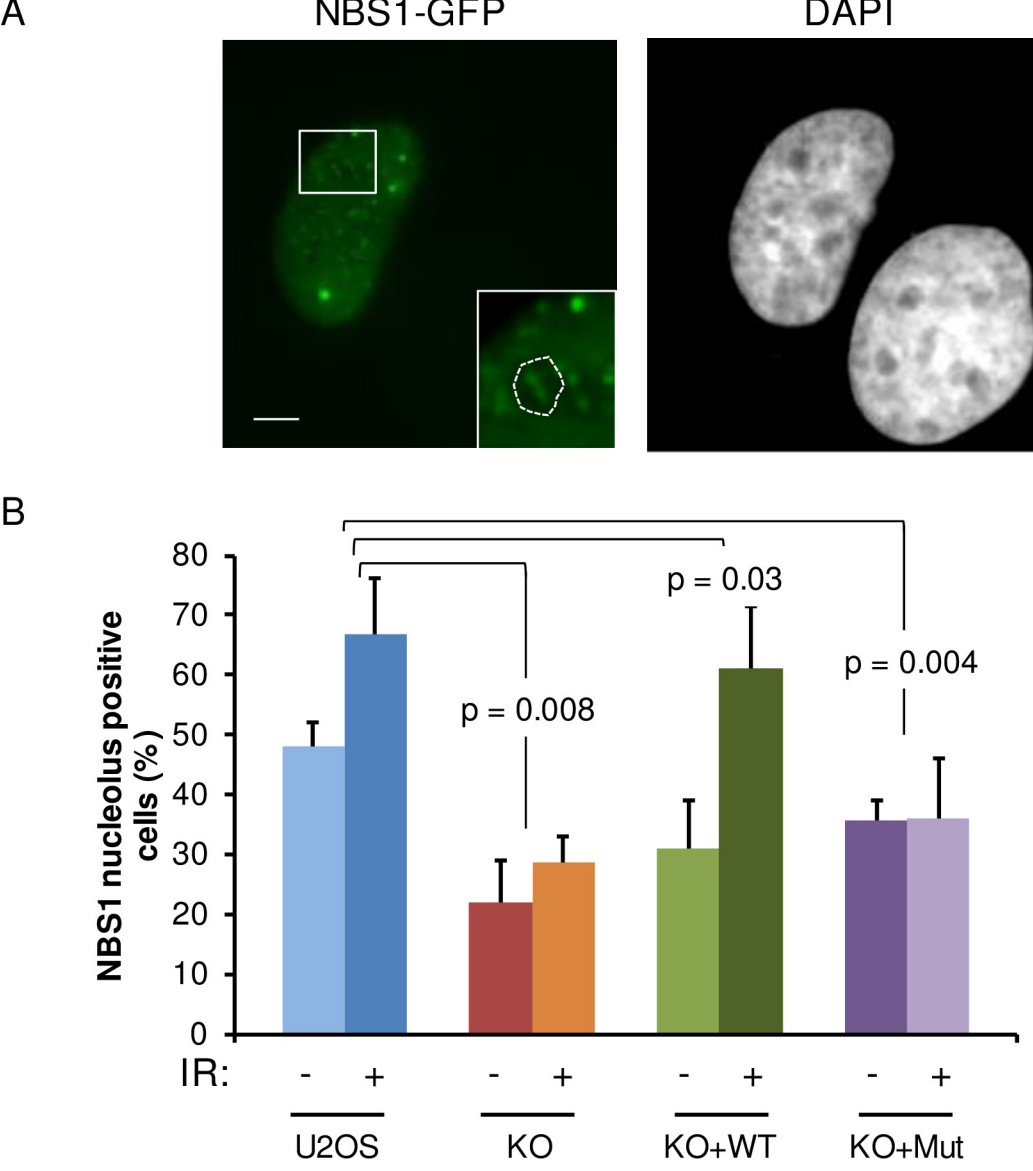

**Fig 10. NBS1 localization in nucleoli after DNA damage is dependent on JMJD6.** (A) Representative images of U2OS cells expressing NBS1-GFP after IR exposure (5 Gy). Insert showing the presence of NBS1 foci in nucleolus. Scale bar 5 μm. (B) Quantification of the percentage of U2OS, JMJD6-KO and JMJD6-KO-complemented with wild type JMJD6 (KO +WT) and JMJD6-KO-complemented with an inactive form of JMJD6 (KO+Mut) cell lines expressing NBS1-GFP that exhibit NBS1 nucleolar foci without and after IR exposure. Results are the mean +/- standard deviation from three independent experiments. In each independent experiment more than 100 transfected cells were counted for each point. The p values of the difference to the U2OS cell line after IR are indicated (Student t test).

Our data suggest that JMJD6 function is intimately linked to the role of Treacle. Indeed, we found a physical interaction between JMJD6 and TCOF1. Moreover, ChIP experiments indicated that JMJD6 is present at rDNA before break induction, as is TCOF1 [28]. Finally, again like TCOF1, we found that JMJD6 is important for the full recruitment of NBS1 in the nucleolus (see our model in Fig 11). Interestingly, in the absence of JMJD6, transcription repression at rDNA still occurs, indicating that there is enough NBS1 to mediate transcriptional repression. However, there are defects of NBS1-dependent processes, such as formation of nucleolar caps.

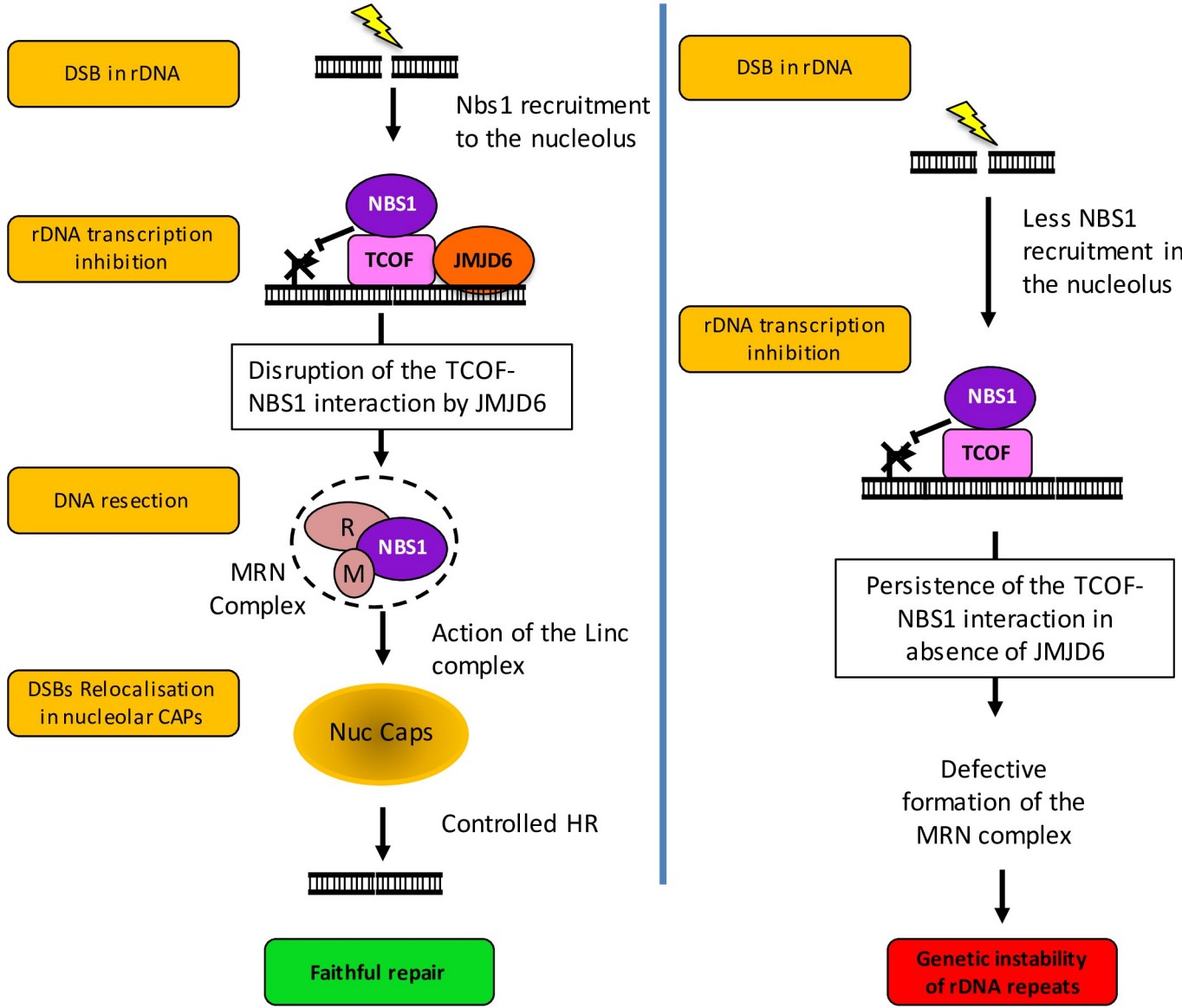

**Fig 11. Working model on the role of JMJD6 in rDNA breaks management.** Upon induction of rDNA breaks in wild type cells (left), JMJD6 in association with TCOF1 restricts the NBS1-TCOF1 interaction, allowing NBS1 to be available to initiate resection within the MRN complex and therefore leading to nucleolar caps formation and faithful homologous recombination. In JMJD6-depleted cells (right), the NBS1/TCOF1 complex still forms, allowing transcriptional repression, but nucleolar caps formation is defective, resulting in a genetic instability of rDNA arrays.

Our data thus suggest that JMJD6 is specifically required for NBS1 to participate in DNA repair processes occurring at the rDNA (Fig 10). Strikingly, we found that JMJD6 restricts the TCOF1-NBS1 interaction but at the same time favours NBS1 recruitment to the nucleolus. What could be the explanation for this apparent discrepancy? Our findings that JMJD6 decreases the interaction between NBS1 and TCOF1 suggests the existence of two pools of

nucleolar NBS1, with JMJD6 controlling the repartition of NBS1 within these two pools. Importantly, recent data show that resection initiated by the MRN complex, in which NBS1 participates, is required for nucleolar caps formation downstream of transcriptional inhibition [12,30]. These results indicate that NBS1 has two separate functions upon DNA damage at rDNA: one in physical association with TCOF1 for mediating transcriptional repression, and the other one in the context of the MRN complex to initiate resection in the HR DNA repair pathway. It is thus tempting to speculate that these two functions of NBS1 could be attributed to the two pools of NBS1 we uncovered. It is possible that following NBS1 mobilization to mediate transcriptional silencing, JMJD6 enzymatic activity is important to restrict the TCOF1-NBS1 interaction, therefore allowing NBS1 to be available for the MRN complex and downstream events. Proteins modifications by JMJD6 could be the molecular events that will initiate repair of DSBs in nucleolar caps once transcription inhibition is achieved. Such a model would additionally explain why JMJD6 catalytic mutant is recruited to breaks at rDNA (via its association with TCOF1) while being defective in allowing NBS1 recruitment to the nucleoli and downstream events.

NBS1 within the MRN complex is involved in the two main repair processes of DSB repair, NHEJ and HR [32]. As discussed above, the MRN complex mediates DNA resection at DSBs, a process which is required for nucleolar caps formation [30], probably explaining why nucleolar caps formation is defective upon JMJD6 inhibition. However, it is also possible that NHEJ is defective upon JMJD6 depletion. Indeed, we found that JMJD6 depletion affects NHEJ in the nucleoplasm, as measured using a reporter substrate stably integrated in DNA but outside the rDNA (Fig 2). Defects into the two main mechanisms of DSB repair in the nucleolus is probably responsible for the genetic instability of rDNA arrays we observed in JMJD6 depleted cells, with a decreased number of rDNA arrays and the appearance of complex rearrangements of repeats (Fig 4).

The fact that actinomycin D treatment induces nucleolar caps led to the proposal that their formation is a direct consequence of transcriptional inhibition [33]. However, we show here an example of a protein whose depletion leads to the uncoupling between transcription inhibition and nucleolar caps formation, indicating that cap formation is an active process. Strikingly, a similar conclusion was drawn from a recent study which shows that the LINC complex is important for nucleolar caps formation downstream of transcriptional inhibition [30]. These observations indicate that the formation of nucleolar caps is not a consequence of transcription inhibition *per se* but requires specific factors. We also found that rDNA transcription is lower upon DNA break induction in JMJD6 depleted cells compared to control cells. This could be due to a direct role of JMJD6 in allowing rDNA transcription recovery following DSB repair by controlling the interaction between Treacle and NBS1. Note however that JMJD6 is not absolutely required for rDNA transcription restart, since we found that 6 hours following irradiation, rDNA transcription is back to normal in JMJD6 knock-down cells (Fig 7). Alternatively, it may be the consequence of a role of JMJD6 in NHEJ-mediated repair of rDNA breaks, since rDNA transcription silencing in response to DNA damage is exacerbated upon inhibition of NHEJ [9]. In addition, our mass spectrometry data reveal an interaction between JMJD6 and PHRF1 in etoposide treated cells (see the complete list of JMJD6 interactors with the link provided in Materials and Methods section). PHRF1 is known to influence DSB repair by NHEJ and to interact with NBS1 [34]. How this would be related to the function of JMJD6 in nucleolar caps formation is unclear and could reflect independent roles of JMJD6 in the management of DNA breaks occurring in rDNA.

Although we studied JMJD6 involvement in the repair of DSBs occurring in rDNA, we cannot rule out that JMJD6 is also involved in the repair of damage occurring at other genomic locations. Strikingly, such an effect was recently described by Huo et al. [23] However, we

obtained results contradictory to them in particular concerning the sensitivity to IR in JMJD6 defective cells. This is probably due to the fact that we examined IR sensitivity in KO cells whereas they used KD performed after siRNA. In addition, the phenotypes they observed are independent of JMJD6 activity, whereas we demonstrate that JMJD6 activity is involved in managements of rDNA breaks. Finally, they identified SIRT1 as interacting with JMJD6 that we do not observe in our mass spectrometry data. Several reasons could explain this latter discrepancy such as the use of overexpressed N-terminus Flag-tagged JMJD6 in Huo et al. study whereas we use endogenously C-terminus Flag-Strep tagged JMJD6. Alternatively, the interaction of SIRT1 is not stable enough or is not conserved in K562 cells in which we performed the purifications of JMJD6-associated proteins. Whether JMJD6 plays a general role in DNA damage response and/or repair, or whether it is important for repair of specific loci clearly merits further investigations. Potential candidate loci could be DNA elements with features similar to rDNA arrays, with a repetitive nature and requiring sequence conservation. In summary, we identified here a new factor participating in the maintenance of rDNA integrity. The importance of such mechanism is highlighted by the increasing number of studies showing the involvement of rDNA repeats integrity in a number of diseases such as cancer, neurological- and aging-associated diseases most of them in relationship with factors associated with the genome maintenance [4,8].

## Materials and methods

### Cell culture conditions

U2OS cell line, and subclones were cultured in DMEM medium supplemented with 10% foetal calf serum, sodium pyruvate and antibiotics (penicillin/streptomycin) in a humidified atmosphere with 5% $CO_2$. K562 cell line was cultured in RPMI 1640 medium supplemented with newborncalf serum. Flp-In T-REx HEK293 cell line was cultured in DMEM medium supplemented with 5% foetal calf serum, 5% cosmic calf serum and antibiotics (penicillin/streptomycin) in a humidified atmosphere with 5% $CO_2$.

### Generation of JMJD6 deficient and tagged cell lines

U2OS cells were made defective for JMJD6 by using CRISPR technology as described in [24]. Homozygote knocked out cell lines were checked by sequencing and Western blot for JMJD6 expression. A KO cell line was transfected with a plasmid coding for a V5-tagged form of JMJD6 (KO+WT) or a JMJD6 catalytically inactive form (H187A-D189A-H273A) (KO+Mut) (gift of Dr M Le Romancer, Lyon, France) and stable clones selected in order to complement the JMJD6 deficiency. A FLAG-Strep tag was inserted in K562 cells at the C-terminus of JMJD6 using CRISPR technology as described [24,35]. For the BioID experiments HEK293 cells were used and constructs for the genes of interest were generated via Gateway cloning into pDEST 3' BirA*-FLAG-pcDNA5-FRT-TO as per [36] for JMJD6 (NM_001081461.1) and TCOF1 (NM_001135243).

### High throughput microscopy

U2OS cells (7500) were transfected with 10nM siRNA using INTERFERIN (Ozyme) according to the manufacturer's instructions and seeded in 96 well plates (Perkin Elmer). Two days after transfection cells were exposed to ionizing radiation (8 Gy) then fixed with formaldehyde (4% in PBS). Cells were permeabilized with triton X100 (0.5% in PBS) for 5 min, stained with stained with γH2AX antibody (1/500) diluted in PBS-BSA 1% and Alexa Fluor 488 anti mouse (Thermofisher). Acquisition was performed on at least 1000 cells per well (3 wells per

condition) with 20X objective with Harmony Imaging Software 4.1 (Perkin Elmer). Image analysis was pursued using Colombus 2.5.0 software (Perkin Elmer) to quantify γH2AX spot intensity.

## 5FUrd incorporation measurement

Transcription of rDNA was monitored by revealing the incorporation of 5-Fluoro-Uridine in Nascent RNA (5-FUrd, Sigma). Cells were incubated in presence of 2mM 5FUrd for 20 minutes followed by fixation in 4% FA (Sigma). They were permeabilized with triton X100 (0.5% in PBS) for 5 min, then stained using an anti BrdU antibody (Sigma, B2531) and Alexa Fluor 488 anti mouse (Thermofisher) diluted 1/500 in PBS-BSA 1% and 0.4 U.mL$^{-1}$ RNAsin (Promega). We visualized and quantified 5-FUrd incorporation by using high content imaging device (OPERETTA, Perkin Elmer). All imunofluorescence steps were performed at 4˚C. Image analysis was pursued using colombus software to measure 5-FUrd spot intensity.

## Laser-induced DNA damage on living cells

The system used to perform laser-induced DNA damage has been previously described in details in [37]. Briefly, the system is composed of an inverted microscope (DMI6000B; Leica) equipped with a temperature controller and a CO2 flow system. DNA damage was generated on nucleus with a green pulsed laser (532 nm). The beam was focused with a 100x NA 1.4 immersion objective (Leica). Images were acquired with a cooled charge-coupled device camera (CoolSNAP HQ2). The system was driven by Metamorph software. U2OS cells were transfected with plasmid expressing JMJD6-GFP in a 2-well chamber (Labtek) in 1 ml of OptiMEM medium without red phenol. Images were recorded using the Metamorph software package (MDS Analytical Technologies).

## Detection of nucleolar caps

DIvA cells were seeded on glass coverslips, transfected with the different siRNA (10nM final concentration) according to the manufacturer's instructions with Jet PEI (Ozyme, France). Two days later, cells were treated with hydroxy-tamoxifen during 4h to generate DNA DSB [29]. Cells were fixed with formaldehyde (3.7% in PBS) permeabilized with triton X100 (0.5% in PBS) for 5 min, washed, then non-specific binding saturated with PBS-BSA (3%) for 1h and incubated with anti UBF antibody (1/500) in PBS-BSA 0.5% for 1h then with secondary antibody coupled to fluorophore (alexa 488) and stained with DAPI. In the case of ionizing radiation exposure, cells were treated with 20 Gy and fixed 6h post IR exposure. When necessary cells were treated with the ATM inhibitor (KU55933) (10 μM final concentration) 1h prior to IR exposure.

## Proximity Ligation Assay (PLA)

The cells were seeded onto glass coverslips, treated (5 Gy) fixed (1h post IR) with formaldehyde (3.7% in PBS) and the PLA assays (Duolink PLA technology, Sigma-Aldrich, DUO92014) performed using manufacturer's protocol with antibody staining performed as in the standard immunofluorescence procedure with antibodies against V5 tag and Treacle or NBS1 and Treacle.

## Imaging

Images were collected with a microscope (DM5000; Leica) equipped with a charge-coupled device camera (CoolSNAP ES; Roper Scientific) and a 100x objective (HCX PI APO ON:1.4–

0.7) and SEMROCK filters. Acquisition software and image processing used the MetaMorph software package (Molecular Devices). Confocal imaging was performed on LSM880 microscope (Zeiss), Z-stacks of fluorescent images were captured and analyzed using ZEN software with a 63x immersion oil objective (Plan-Apochromat ON:1.4).

## Immublotting detection

Whole cell extracts were prepared in Laemmli buffer (Tris HCl 62.5mM, SDS1%, Mercaptoethanol 5%, glycerol 25% and bromophenol blue). Samples were sonicated and heated at 95˚C for 5 minutes before their separation on a 4–15% gradient SDS PAGE gel (Biorad, 4–15%). Proteins were transferred from gel to nitrocellulose membrane using semi-dry method. Once blocked in PBS-Tween (5%) plus 10% skimmed milk, membrane were firstly immunoblotted with primary antibody diluted in 2% milk PBS-T (generally 1/1000 except for anti H3 whose dilution used is 1/10000) at 4˚C overnight and finally immunobloted with secondary antibody coupled to HRP (Sigma) diluted in 2% milk PBS-T at room-temperature for 1 hour. After several washes in PBS-T, proteins were detected using ECL lumi-light plus (Roche) and images acquired using camera system (Chemitouch Biorad).

## Immunoprecipitations

Whole cell extracts were diluted in RIPA buffer and precleared with protein A beads (1h at 4˚C). Immunoprecipitations were performed by adding 1 µg of antibodies to precleared extracts (4˚C overnight). A mixture of protein A/G beads were added for 1h at 4˚C. Beads were washed three times with RIPA buffer and subjected to Western blotting.

## Clonogenic assay

Cells were seeded at 500 cells per well in 6 well plates and let to attach overnight. Plates were irradiated using a Biobeam 8000 ([137]Cs source) (Anexplo service, Rangueil, Toulouse, France). Plates were kept in cell incubator for 10 days then stained with crystal violet. Colonies of more than 50 cells were counted.

For purpose of testing cell sensitivity to DSB in rDNA, U2OS cells (1 x 10[6]) were transfected with siRNA (1µM) by electroporation (4D-Nucleofactor, Amaxa) and seeded at 1.10[5] cells per well in 6-well plates. 48h after transfection, cells were transfected by JetPEI (Ozyme) according to manufacturer's instructions with either an empty px330 plasmid (Addgene) coding only for Cas9 or a px330 plasmid coding for Cas9 and a sgRNA targeting rDNA [13], together with px330 coding for a sgRNA targeting ATP1A1 gene together with a donor plasmid to generate a mutated ATP1A1 gene coding for an ouabain insensitive Na$^+$/K$^+$ ATPase [24]. 48 hours after plasmids transfection, ouabain (0.7 µM, Sigma, France) was added to the culture medium to select transfected cells in which CRISPR-induced cleavage and mutation insertion happened. Two weeks later cells were fixed and stained with crystal violet solution and the colonies counted.

## Affinity Purification and mass spectrometry analysis of endogenous JMJD6 interactome

K562 cells expressing 3xFLAG-2xStrep tag at the C-terminus of endogenous JMJD6 were amplified (1.5 x10[9] cells), nuclear cell extracts prepared and used to perform tandem affinity purification as described in [38].

Briefly, nuclear extracts [39] were adjusted to 0.1% Tween-20, and ultracentrifuged at 100,000 g for 45 min. Extracts were precleared, then 250 ul anti-FLAG M2 affinity resin

(Sigma) was added for 2 hr at 4°C. The beads were then washed in Poly-Prep columns (Bio-Rad) buffer #1 (20 mM HEPES-KOH[pH 7.9], 10% glycerol, 300 mM KCl, 0.1% Tween 20, 1 mM DTT, Halt protease and phosphatase inhibitor cocktail [Pierce]) followed buffer #2 (20 mM HEPES-KOH [pH 7.9], 10% glycerol, 150 mM KCl, 0.1% Tween 20, 1mMDTT, Halt protease and phosphatase inhibitor cocktail [Pierce]). Complexes were eluted with buffer #2 supplemented with 150 ug/ml 3xFLAG peptide (Sigma) for 1 hr at 4°C. Next, fractions were mixed with 125 ul Strep-Tactin Sepharose (IBA) affinity matrix for 1 hr at 4°C, and the beads were washed with buffer #2 in Poly- Prep columns (Bio-Rad). Complexes were eluted in two fractions with buffer #2 supplemented with 2.5mM D-biotin, flash frozen in liquid nitrogen, and stored at -80°C. Typically, 15 ul of the first elution (3% of total) was loaded on NuPAGE 4%–12% Bis-Tris gels (Life Technologies) and analyzed by silver staining.

For mass spectrometry analysis, fractions were loaded on gel and migrated for about 1 cm, then stained with sypro ruby red and a gel slice containing the entire fraction was cut for in-gel trypsin digestion and analysis on a LC-MS/MS apparatus (Thermo scientific Orbitrap Fusion) at the Proteomics Platform of the Quebec Genomics Center.

## Proximity biotinylation assay (BioID):

Tetracycline-inducible HEK293 Flp-In T-REx cells stably expressing BirA*-FLAG-JMJD6 or TCOF1 fusion protein were grown in 15 cm plates to 75% confluence, two plates per replicate. Negative controls samples for BioID experiments were parental Flp-In T-REx HEK293 stable cells expressing BirA*-FLAG fused a green fluorescent protein (GFP) as per [36]. The cells were treated with 1 μg/μL of tetracycline for 20 hours and subsequently with 50 μM biotin for 4 hours while being incubated at 37°C. Cells were concurrently treated with 20 uM etoposide, or DMSO, with the biotin incubation. After treatments, cells were washed with fresh 1X PBS and were lysed in 1.5 mL of RIPA lysis Buffer on ice. RIPA lysis buffer: 1% (v/v) NP-40, 0.1% SDS, 50 mM Tris-HCl pH7.4, 150 mM NaCl, 0.5% (w/v) Sodium Deoxycholate, 1 mM EDTA; supplemented with 1 mM DTT, 1mM PMSF (Bio Basic INC, PB0425) and 1X protease inhibitor cocktail (SIGMA, P8340) prior to utilization. Samples were sonicated to shear the chromatin (30 secs at power ~4 using Sonic Dismembrator 60 equipped with 1/8" probe) on ice. Further, 250 units of turbonuclease were added and incubated on rotator for 1 hour at 4°C. The sample was centrifuged at 14,000 rpm for 20 min at 4°C.

## Streptavidin-based affinity capture of biotinylated proteins:

The Streptavidin Sepharose beads were washed twice in 1 mL of lysis buffer (60 μL of slurry per sample). The beads were pelleted by centrifugation after each washing and were then incubated with samples at 4°C on rotator for 3 hours. The bound beads were pelleted by centrifugation and were washed twice with 1 mL of RIPA lysis buffer (without protease inhibitors) and transferred to a new 1.5 mL Eppendorf tube to minimize background contaminants. Tubes were centrifuged and the supernatants were discarded. The beads were washed 3 times with 1 mL of 50 mM Ammonium bicarbonate (ABC).

## Sample preparation for LC-MS/MS analysis:

On-beads trypsin digestion: the beads were resuspended in 100 μL of 50 mM ABC and with 1μg of trypsin (resuspend in Tris-HCl pH 8.0). The samples were incubated overnight (~15 hours) at 37°C with shaking with an extra 1μg of trypsin added subsequently for 2–4 hours to insure complete digestion. Samples were gently centrifuged and the supernatant transferred in new tubes. The beads were rinsed twice and the supernatant pooled.

## Peptides recovery and desalting

The peptide digestion was stopped by adding formic acid (from a 50% stock solution) to a final concentration of 2%. Samples were dried in a Speed-Vac vacuum concentrator without heat. The samples were desalted with $C_{18}$ StageTips and sent for MS processing. The $C_{18}$ StageTips were prepared according to the published protocol by Rappsilber et al.[40]. Conditioning: The disks were made wet by passing 20 μL of 100% methanol through the StageTip. Then, 20 μL of buffer B (0.5% formic acid, 80% acetonitrile in water) was added to the StageTip and centrifuged (3000 rpm, 30 secs). Further, 20 μL of buffer A (0.5% formic acid in water) was added to the StageTip and centrifuged (3000 rpm, 30 secs). The samples prepared for AP were directly loaded to the $C_{18}$-StageTips. For BioID, the dried samples were resuspended in 20μl of buffer A before loading to the $C_{18}$-StageTips. The tips were centrifuged at 3000 rpm for 3 min. The $C_{18}$-StageTips was washed twice with 20μl of buffer A at 3000 rpm for 3 min each time. The sample was eluted by placing the $C_{18}$-StageTips in a fresh tube and adding 20μl of buffer B. The tubes were centrifuged at 3000 rpm for 30sec. The process was repeated 3 times. The elute was dried using SpeedVac and sent for LC-MS/MS analysis.

## Proteins identification by mass spectrometry

The analyses were performed at the proteomic platform of the Quebec Genomics Center. Peptide samples were separated by online reversed-phase (RP) nanoscale capillary liquid chromatography (nanoLC) and analyzed by electrospray mass spectrometry (ESI MS/MS). The experiments were performed with a Dionex UltiMate 3000 nanoRSLC chromatography system (Thermo Fisher Scientific) connected to an Orbitrap Fusion mass spectrometer (Thermo Fisher Scientific) equipped with a nanoelectrospray ion source. Peptides were trapped at 20 ul / min in loading solvent (2% acetonitrile, 0.05% TFA) on a 5mm x 300 μm $C_{18}$ pepmap cartridge pre-column (Thermo Fisher Scientific) during 5 minutes. Then, the pre-column was switch online with a self-made 50 cm x 75 um internal diameter separation column packed with ReproSil-Pur $C_{18}$-AQ 3-μm resin (Dr. Maisch HPLC) and the peptides were eluted with a linear gradient from 5–40% solvent B (A: 0,1% formic acid, B: 80% acetonitrile, 0.1% formic acid) in 90 minutes, at 300 nL/min. Mass spectra were acquired using a data dependent acquisition mode using Thermo XCalibur software version 3.0.63. Full scan mass spectra (350 to 1800m/z) were acquired in the orbitrap using an AGC target of 4e5, a maximum injection time of 50 ms and a resolution of 120 000. Internal calibration using lock mass on the m/z 445.12003 siloxane ion was used. Each MS scan was followed by acquisition of fragmentation spectra of the most intense ions for a total cycle time of 3 seconds (top speed mode). The selected ions were isolated using the quadrupole analyzer in a window of 1.6 m/z and fragmented by Higher energy Collision-induced Dissociation (HCD) with 35% of collision energy. The resulting fragments were detected by the linear ion trap in rapid scan rate with an AGC target of 1E4 and a maximum injection time of 50ms. Dynamic exclusion of previously fragmented peptides was set for a period of 20 sec and a tolerance of 10 ppm.

## Data Dependent Acquisition MS analysis

Mass spectrometry data was stored, searched and analyzed using the ProHits laboratory information management system (LIMS) platform [41]. Thermo Fisher scientific RAW mass spectrometry files were converted to mzML and mzXML using ProteoWizard (3.0.4468;[42]. The mzML and mzXML files were then searched using Mascot (v2.3.02) and Comet (v2012.02 rev.0). The spectra were searched with the RefSeq database (version 57, January 30th, 2013) acquired from NCBI against a total of 72,482 human and adenovirus sequences supplemented with "common contaminants" from the Max Planck Institute (http://141.61.102.106:8080/

share.cgi?ssid=0f2gfuB) and the Global Proteome Machine (GPM; http://www.thegpm.org/crap/index.html). Charges +2, +3 and +4 were allowed and the parent mass tolerance was set at 12 ppm while the fragment bin tolerance was set at 0.6 amu. Carbamidomethylation of cysteine was set as a fixed modification. Deamidated asparagine and glutamine and oxidized methionine were allowed as variable modifications. The results from each search engine were analyzed through TPP (the Trans-Proteomic Pipeline (v4.6 OCCUPY rev 3)[43] via the iProphet pipeline [44].

## MS data archiving and availability

All MS files used in this study were deposited at MassIVE (http://massive.ucsd.edu) and at ProteomeXchange (http://www.proteomexchange.org/). They were assigned the identifiers MassIVE MSV000083409 and ProteomeXchange PXD012603.

## Measurement of genetic instability at rDNA

Cells were seeded on glass coverslips and irradiated at 2 Gy and let recover for 24h before adding colcemid (0.1 μg/ml) for 3h to enrich the cell population in metaphases. Then cells were fixed with formaldehyde (3.7% in PBS) permeabilized with Triton X100 (0.5% in PBS) for 5 min, washed, then non-specific binding saturated with PBS-BSA (3%) for 1h and incubated with anti UBF antibody (1/500) in PBS-BSA 0.5% for 1h then with secondary antibody coupled to fluorophore (alexa 488) and stained with DAPI.

## DNA FISH combing on ribosomal DNA

Cells were transfected with control siRNA (siCtrl), or two different siRNAs directed against JMJD6 (siJMJD6-1 and siJMJD6-2) and 48h later induced or not for DSB by treatment with OHTam during 24h. At the end of the treatment cells were harvested and included in low melting agarose and sent to the Genomic Vision Company for combing and FISH. DNA fibers stained as represented in Fig 4 were then detected using web interface provided by Genomic Vision company. Identification and counting of rearrangement events were estimated using a script designed to count the total number of rDNA units and assessing the the occurence of changes when comparing one unit to the next. GitHub https://github.com/LegubeDNAREPAIR/rDNA/blob/master/get_break_event.py

## Chromatin Immunoprecipitation (ChIP)

Cells were double crosslinked with DMA (0.25%, 45 min), washed with PBS then with formaldehyde (1%, 20min) and ChIP was performed as described [45] using 200 μg of chromatin. Briefly, nuclei were prepared and sonicated to obtain DNA fragments of about 500–1000 bp. Following preclearing and blocking steps, samples were incubated overnight at 4°C with specific antibodies (anti V5) or without antibody (mock) as negative control. Immune complexes were then recovered by incubating the samples with blocked protein A/G beads for 2 h at 4°C on a rotating wheel. After extensive washing, crosslink was reversed by adding RNase A to the samples and incubating overnight at 65°C. After a 1h30 proteinase K treatment, DNA was purified with the GFX PCR kit (Amersham), and analysed by Q-PCR on a CFX96 Real-time system device (BioRad) using the IQ SYBR Supermix (BioRad Laboratories, Marnes-la-Coquette, France), according to the manufacturer's instructions. All samples were analyzed in triplicates.

## DSB repair activity measurement

DSB repair activity in response to I-SceI endonuclease was performed as previously described in U2OS cells harboring a substrate designed for recording homology driven DSB repair [46,47] and in immortalized human fibroblasts, GCS5, harboring NHEJ substrate [47,48].

## List of antibodies used

Anti JMJD6 (Santa Cruz, sc28349), anti UBF (Bethyl, A301-859A), anti histone H3 (Abcam, Ab1791), anti BrdU (Sigma, B2531), anti gamma H2AX (Cell signaling, 9718(20E3)), anti GAPDH (Chemicon, MAb374), anti myc (Santa Cruz, sc-40), anti Treacle (Santa Cruz, sc374536), anti V5 (Cell Signaling, 12032), anti NBS1 (Sigma, PLA0179), anti phospho ATM (S1981) (Cell signaling, 10H11.E12).Anti Rad51 (Millipore, PC130), Anti RPA S33 (Bethyl, A306-246A-T)

## List of oligonucleotides used

siRNA control: CTTACGCTGAGTACTTCGA;
 siRNA JMJD6-1: CAGCUAUGGUGAACACCCUAA;
 siRNA JMJD6-2: CCAAAGUUAUCAAGGAAAU
 sgRNA: rDNAtarget: GCCTTCTCTAGCGATCTGAG
 sgRNA: JMJD6KO: GAGCAAGAAGCGCATCCGCG
 sgRNA: JMJD6 tag: CCAGGTGACCCAGCAAGGCT

## Statistical analysis

Data obtained with operetta device were analysed using Wilcoxon Mann-Whitney test. For data with lower number of values a Student t test was performed (clonogenic assay).

## Supporting information

**S1 Fig. Laser induced DNA damage on living cells.**
(PDF)

**S2 Fig. JMJD6 recruitment at DSB assessed by ChIP.**
(PDF)

**S3 Fig. JMJD6 KO cell lines present increased sensitivity to ionizing radiations.**
(PDF)

**S4 Fig. Recruitment of JMJD6 mutant at DNA damages.**
(PDF)

**S5 Fig. JMJD6 colocalizes with Treacle in nucleolus.**
(PDF)

**S6 Fig. TCOF1 expression and localization in JMJD6 KO cells.**
(PDF)

**S7 Fig. JMJD6 influences the NBS1-Treacle interaction.**
(PDF)

**S8 Fig. JMJD6 colocalizes with Treacle and NBS1 in nucleolus.**
(PDF)

**S9 Fig. rDNA transcription in JMJD6-depleted MRC5 cells after induction of rDNA DSB.**
(PDF)

**S10 Fig. Cell cycle analysis of the JMJD6-KO cell line and complemented cell lines.**
(PDF)

**S11 Fig. JMJD6 depletion affects nucleolar caps generation after ionizing radiations exposure.**
(PDF)

**S1 Table. JMJD6 interacting proteins obtained after affinity purification-mass spectrometry (AP-MS)**
(XLSX)

**S2 Table. JMJD6 and TCOF1 interacting proteins obtained after Bio-ID.**
(XLSX)

**S1 Data.** Fig 1.
(XLSX)

**S2 Data.** Fig 2.
(XLSX)

**S3 Data.** Fig 3A
(XLSX)

**S4 Data.** Fig 3C
(XLSX)

**S5 Data.** Fig 4B.
(XLSX)

**S6 Data.** Fig 4E.
(XLSX)

**S7 Data.** Fig 5D.
(XLSX)

**S8 Data.** Fig 6.
(XLSX)

**S9 Data.** Fig 7A.
(XLSX)

**S10 Data.** Fig 7B.
(XLSX)

**S11 Data.** Fig 8B.
(XLSX)

**S12 Data.** Fig 8D and 8E.
(XLSX)

**S13 Data.** Fig 9.
(XLSX)

**S14 Data.** Fig 10.
(XLSX)

**S15 Data. S1E Fig.**
(XLSX)

**S16 Data. S2 Fig.**
(XLSX)

**S17 Data. S3 Fig.**
(XLSX)

**S18 Data. S7 Fig.**
(XLSX)

**S19 Data. S10 Fig.**
(XLSX)

**S20 Data. S11 Fig.**
(XLSX)

## Acknowledgments

We thank Thomas Mangeat for his assistance with the laser track experiments and Magali Suzanne for assistance with the confocal experiments. We are grateful to the Non-Invasive Exploration service- US006/CREFRE INSERM/UPS for giving us the access to the irradiator Biobeam 8000 (Rangueil, Toulouse). We acknowledge the Toulouse Regional Imaging—Light Imaging Toulouse CBI Platform for technical assistance in high content microscopy. We thank Florence Larminat for her critical reading of the manuscript.

## Author Contributions

**Conceptualization:** Jacques Côté, Didier Trouche, Yvan Canitrot.

**Formal analysis:** Jérémie Fages, Catherine Chailleux, Jacques Côté, Didier Trouche, Yvan Canitrot.

**Funding acquisition:** Jean-Philippe Lambert, Jacques Côté, Didier Trouche, Yvan Canitrot.

**Investigation:** Jérémie Fages, Catherine Chailleux, Jonathan Humbert, Suk-Min Jang, Jérémy Loehr, Jean-Philippe Lambert, Yvan Canitrot.

**Methodology:** Yvan Canitrot.

**Project administration:** Didier Trouche.

**Supervision:** Jean-Philippe Lambert, Jacques Côté, Didier Trouche, Yvan Canitrot.

**Validation:** Jean-Philippe Lambert, Didier Trouche, Yvan Canitrot.

**Visualization:** Yvan Canitrot.

**Writing – original draft:** Didier Trouche, Yvan Canitrot.

**Writing – review & editing:** Jacques Côté, Didier Trouche, Yvan Canitrot.

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
