## [Decision Letter · Decision Letter 0]

5 Dec 2019

Dear Dr canitrot,

Thank you very much for submitting your Research Article entitled 'JMJD6 participates in the maintenance of ribosomal DNA integrity in response to DNA damage' to PLOS Genetics. Your manuscript was fully evaluated at the editorial level and by independent peer reviewers. The reviewers appreciated the attention to an important problem, but each of them raised substantial concerns about the current manuscript. Based on the reviews, we will not be able to accept this version of the manuscript, but we would be willing to review again a much-revised version. We cannot, of course, promise publication at that time.

The reviewers have provided extensive comments on the manuscript. Should you decide to revise the manuscript for further consideration here, your revisions should address the specific points made by each reviewer. We will also require a detailed list of your responses to the review comments and a description of the changes you have made in the manuscript.

If you decide to revise the manuscript for further consideration at PLOS Genetics, please aim to resubmit within the next 60 days, unless it will take extra time to address the concerns of the reviewers, in which case we would appreciate an expected resubmission date by email to plosgenetics@plos.org.

[LINK]

We are sorry that we cannot be more positive about your manuscript at this stage. Please do not hesitate to contact us if you have any concerns or questions.

Yours sincerely,

Nancy Maizels, Ph.D.

Associate Editor

PLOS Genetics

Gregory Barsh

Editor-in-Chief

PLOS Genetics

Reviewer's Responses to Questions

**Comments to the Authors:**

Reviewer #1: Review report attached.

Reviewer #2: Fages et al., identify JMJD6 as a protein involved in nucleolar DNA repair. At first the authors perform a screen to identify proteins, that upon depletion, elevate gH2AX levels after Ionising radiation. From this screen they select JMJD6 as a potential repair protein candidate for further investigation. Micro-laser irradiation reveals a strong consistent recruitment to nucleoli after DNA damage and a more moderate recruitment in nuclear chromatin. Clonogenic survival assay show increased sensitivity to nuclear and nucleolar DNA damage after depletion of JMJD6. rDNA instability is documented by decrease in mitotic UBF foci number and rearrangement of rDNA repeats. The authors then identify an interaction with the nucleolar protein Treacle/TCOF1 and that depletion of JMJD6 increases the interaction between Treacle and NBS1. The authors also find that transcriptional inhibition is not compromised in the absence of JMJD6. The authors then report that nucleolar cap formation is decreased in JMJD6 depleted cells and suggest that nucleolar cap formation is a process uncoupled from transcriptional inhibition. Finally the authors report less nucleolar NBS1 accumulation after IR in JMJD6 KO cells and propose that JMJD6 is responsible for NBS1 recruitment into nucleoli in an analogous way to Treacle/TCOF1. The data presented in this manuscript are interesting but in some places the experimental design makes it difficult to determine if the conclusions are valid. Furthermore, in some cases additional data is required to support the presented experiments.

Specific comments for the authors

Introduction:

Page 5 line 4: When describing the nucleolar DNA damage response the novel discoveries of the involvements of ATR in nucleolar cap formation should be mentioned (Velichko et al., and Korsholm et al.,).

Figure 2:

Page 8 line 1: “Considering the size of the rDNA loci, the irradiation dose we used in the clonogenic cell survival assay should not lead to breaks in the rDNA in a significant proportion of cells”. The authors should explain how this was calculated and include references that documents this statement.

Figure 3E

The authors state “Results show that in absence of external DNA damage JMJD6 depletion

per se induced a higher level of rDNA rearrangements (Figure 3E and Figure S5). In response to induced-DSB we observed an increase in rDNA rearrangements in control cells which was higher in JMJD6-depleted cells.” The data presented do not support the that induced-DSBs lead to more rearrangements in JMJD6 depleted cells compared to control cells. On the contrary the level of rearrangements induced by DSBs appears to be less or the same. The authors should provide a third experiment and test if the rearrangements induced with or without damage are significant.

Figure 5:

The authors report an increased interaction between NBS1 and TCOF1 in JMJD6-KO cells. Does this interaction occur in the nucleoli? The authors must co-stain with a nucleolar marker and quantify NBS1-TCOF1 PLA foci in the nucleus versus nucleoli to determine if this interaction is linked to nucleolar functions.

Figure 7:

Page 12, the authors state “However, here we show that nucleolar caps formation and

transcription inhibition can in fact be uncoupled since JMJD6 depleted cells display

transcriptional repression, with yet less nucleolar caps.” If the authors want to claim uncoupling of nucleolar transcription and cap formation they must conduct experiments where these two read outs are measured at the same time with exposure to the same treatments. The current data is not consistent enough to make such claims.

Figure 9:

The authors find that NBS1 recruitment to nucleoli is compromised in JMJD6 KO cells and assigns a role for JMJD6 in NBS1 recruitment. Previous studies, also cited in this paper, have shown Treacle/TCOF1 to be responsible for NBS1 recruitment into nucleoli. Furthermore, the authors have demonstrated that TCOF1-NBS1 interactions are increased in Figure 5 and it is unclear if this occurs in the nucleolus. An alternative explanation could therefore be an alteration in the level/localization of TCOF1 due to the significant amount of nuclear DNA damage generated in the absence of JMJD6. The authors should assay the total level of TCOF1 protein in the cell and the cellular localization of TCOF1 by IF to exclude that the observed changes in NBS1 recruitment is caused by altered localization of TCOF1. Finally the authors should test if NBS1 and TCOF1 co-localize in JMJD6 KO cells.

General comments:

Major concern:

The changes and uncpecific action of the DNA damaging agents used throughout the manuscript makes it difficult to distinguish the nucleolar response from a general stress response activated in the nucleus. Absence of JMJD6 induces DNA damage genome-wide and the nuclear response to exogenous DNA damage is elevated upon treatment with IR. The authors repeatedly uses methods that activates multiple responses (DIvA endonuclease and IR) and a specific role for JMJD6 in the nucleolus therefore becomes hard to determine.

Minor comments:

In all pictures, where relevant, the nucleolus should be indicated either by co-staining with a nucleolar marker or by an added line (generated in a computer software) following the edge of the nucleolus. Currently it is difficult to determine the nuclear localization of features such as caps and foci in the included images and therefore it is difficult to judge their link to nucleolar functions.

The dosis of IR should be stated in all the figure legends.

Reviewer #3: The authors identified JMJD6 in a knock down screen for histone demethylases in the DDR. JMJD6 localises to both nuclear and nucleolar sites of DNA damage using the ‘laser striping assay. However, chromatin immunoprecipitation studies revealed that JMJD6 is already bound to rDNA and not further enriched after induction of an AsiSI-induced DSB within the rDNA (note that AsiSI cuts at about 100 sites throughout the genome in addition to a cut site in rDNA). The authors suggest that JMJD6 at uncut sites could mask recruitment of AsiSI if it only cuts a few rDNA repeats. This is possible but the authors do not provide evidence to support this possibility and the discrepancy between the ‘laser striping’ and ChIP assay with respect to JMJD6 enrichment at sites of DNA damage is left open. Survival assays demonstrate that the catalytic activity of JMJD6 is required for survival after ionising radiation, although recruitment to DNA damage induced by ‘laser striping’ was not dependent upon catalytic activity. To assess DSBs specifically in rDNA the authors nicely used a previously described CRISPR/Cas9 and gRNA targeting rDNA (van Sluis & McStay 2015 G&D) and an innovative co-selection strategy published by other (Agudelo et al 2017, Nat Meths) to assess survival upon DSBs induced specifically in rDNA repeats. They observed that JMJD6 depletion resulted in decreased survival to rDNA-specific DSBs. Importantly, JMJD6 is important for the integrity of the rDNA repeats as the number of Nucleolar Organising Regions (NORs, 10 corresponding to each of the 5 acrocentric chromosomes) detected in metaphase is reduced upon loss of JMJD6 and further reduced upon IR treatment. Ribosomal DNA rearrangements where directly observed by FISH and shown to be dependent upon JMJD6 both before and after inductions of genome-wide DSBs (IR). To address mechanism the authors first performed a proteomic screen to identify JMJD6 partner proteins. In addition to protein involved in splicing, a known function for JMJD6, the authors identified TCOF1 (Treacle) a nucleolar protein already known to be involved in the nucleolus-specific DDR, specifically, it is required for the localisation of NBS1 into the nucleolus in response to DNA damage for repression of rDNA transcription. Relatively weak data is presented suggesting that JMJD6 negatively regulates the interaction between Treacle and NBS1. Given the role of Treacle and NBS1 in the regulation of transcriptional repression upon DNA damage the authors assessed the impact of JMJD6 upon this function using the AsiSI system, as well as IR, both of which result in DSBs throughout the nucleus, as well as the rDNA repeats. The authors interpretation of their data is that JMJD6 does not affect the rDNA transcriptional response to DNA damage, although they confusingly state that: “strikingly, rDNA transcription was further decreased in JMJD6-depleted cells compared to control cells”. Whether JMJD6 has a modulatory role on the rDNA transcriptional response to DNA damage, consistent with a modulatory role on the Treacle-NBS1 interaction known to be required for this response, will require further experimentation (see below). However, the authors report a significant defect in the formation of nucleolar caps upon JMJD6 depletion, suggesting that this response can be uncoupled from rDNA transcriptional repression upon DNA damage. Furthermore, JMJD6 also appears to play a significant role in regulating the localisation of NBS1 into the nucleolus.

While the manuscript as presented has novelty, many results require significant further validation and new approaches (detailed below). The manuscript, at present, is not suitable for publication.

Figure 1. Correct the labelling and legend of Fig 1. Legend indicates that part A is the western, part B is the IF images and C is the quantification. Why does siJMJD6-2 not produce a similar result to siJMJD6-1 in the �H2AX IF, despite the quantification suggesting even greater �H2AX intensity? Correct or comment. Also comment that the nuclear and nucleolar striping of JMJD6-GFP in supplemental Figure S1C have similar timing indicating that recruitment to these two compartments is coincident in those cells that display JMJD6-GFP localisation to laser stipes in both compartments. Could this be an artefact of fluctuations of laser power between different experiments or can this result be seen in different cells from the same experiment? As the authors failed to see increased recruitment of JMJD6 via ChIP (Fig S2), confirmation of JMJD6 recruitment to nucleolar-specific DSBs induced via CRISPR/Cas9 should be attempted to resolve the issue of whether this protein is actually recruited to DSBs, i.e. confirm the laser striping result, especially as this technique is prone to artefact.

Figure 3E. A third experiment with proper quantification and statistical significance should be included. This is important as the effect sizes are relatively small. Also, Fig S5. Would a knock out give better results than the two knock downs used?

Figure 4. For the convenience of the reader, the complete results of the proteomic screen should be provided in a supplemental table, as well as in the MassIVE database. Validation of the JMJD6 interaction with Treacle by PLA and co-localisation is insufficient. The authors should also use another biochemical approach, e.g. co-IP. Also, is this interaction direct (detectable with recombinant proteins) or indirect? The legend for Figure 4 require more information about the experiment shown to facilitate the reader.

Figure 5. Again the effects of JMJD6 on the Treacle-NBS1 interaction must be confirmed biochemically (e.g. co-IP) as the PLA assay is insufficient for the conclusions drawn. A third repeat with proper quantification is also required, especially important given the relatively small effect sizes shown. The legend to his figure also requires more information to facilitate the reader.

Figure 6. As noted above, there is confusion in the manuscript between an observation noted as: “strikingly, rDNA transcription was further decreased in JMJD6-depleted cells…” and their interpretation of no effect on the rDNA transcriptional response. Why did the authors not use the more specific CRISPR/Cas9 approach, as used in Figure 2, to induced DSBs specifically in rDNA repeats? As this would be free of any confounding effects of DSBs elsewhere in the nucleolus it may well give more specific results with perhaps greater effect sizes. While I think the data is clear that JMJD6 does not have a large role in the rDNA transcriptional role to DNA damage, if it does modulate the Treacle-NBS1 interaction it might be expected to modulate this transcriptional response. Can the authors reassess the possible role of JMJD6 in the rDNA transcriptional response to DNA damage specific to the rDNA repeats? They have this capability as demonstrated in Figure 2 and with the addition of a transfection control this approach should be possible. Once more the JMJD6-specific effect sizes of the experiments shown are, unfortunately, small (certainly not striking!). Therefore, quantification of three separate experiments should be combined and analysed for significance as a combined data set. Also, use the JMJD6 knockout rather than siRNA-mediated depletion. The legend to his figure too requires more experimental information to facilitate the reader. Replace “OHT” with “AsiSI”.

Figure 7. This is a convincing result suggesting uncoupling of rDNA transcriptional repression from nucleolar cap formation. However, it would be more convincing to include an experiment using the CRISPR/Cas9-induced rDNA-specific DSBs. Note Figure S10, using IR rather than AsiSI, is missing from the downloaded PDF.

Figure 8. If JMJD6 is specific to the nucleolar DDR using IR is a poor choice of DSB-inducing agent for these experiments and should be repeated with rDNA-specific DSBs.

Figure 9. In part A, the images are not convincing, as shown, with respect to NBS-GFP foci being actually in a nucleolus. It would be much better resolution to show both the DAPI and GFP channels in black and white (similarly for other Figures), rather than false coloured green and blue. Once more, repeating with rDNA-specific DSBs would add significantly to this figure and increase confidence in the overall result.

**Have all data underlying the figures and results presented in the manuscript been provided?**

Reviewer #1: Yes

Reviewer #2: Yes

Reviewer #3: No: Fig S10 was omitted from the supplementary data PDF

PLOS authors have the option to publish the peer review history of their article (what does this mean?). If published, this will include your full peer review and any attached files.

Reviewer #1: No

Reviewer #2: No

Reviewer #3: No

---

## [Decision Letter · Decision Letter 1]

9 Apr 2020

Dear Dr canitrot,

Thank you very much for submitting your Research Article entitled 'JMJD6 participates in the maintenance of ribosomal DNA integrity in response to DNA damage' to PLOS Genetics. Your manuscript was fully evaluated at the editorial level and by independent peer reviewers. The reviewers appreciated the attention to an important topic but identified some aspects of the manuscript that should be improved.

We therefore ask you to modify the manuscript according to the review recommendations before we can consider your manuscript for acceptance. Your revisions should address the specific points made by each reviewer.

We would normally hope to receive your revised manuscript within the next 30 days, but we know that it might take longer in this unusually challenging time.  If you anticipate difficulties in returning it in less than60 days, we would ask you to let us know the expected resubmission date by email to plosgenetics@plos.org.

[LINK]

Yours sincerely,

Nancy Maizels, Ph.D.

Associate Editor

PLOS Genetics

Gregory Barsh

Editor-in-Chief

PLOS Genetics

Reviewer's Responses to Questions

**Comments to the Authors:**

Reviewer #1: The additional experiments performed by the authors have improved the quality of manuscript, as it now contains supportive evidence for the most important observations.

However, the authors should provide further explanation and insight in the discussion on the results with the catalytic mutant (Fig 3A, S4, 6 and S7), which seem contradicting.

Although little mechanistic insight is revealed an explanatory model could be helpful for readers. Also, for the sake of the reader, further improvements to the outline of the manuscript and formatting of the figures are still urgently advised.

Reviewer #2: The manuscript by Fages et al. has improved as a result of several new data panels and text edits. One of the major points of interest, the uncoupling of transcriptional inhibition and cap formation, has also been strengthened in the current version of the manuscript. However, a few points still need to be addressed as outlined below.

Comments to the authors:

“We also examined JMJD6-GFP recruitment after specific induction of rDNA DSB using CRISPR-Cas9. We could observe some cells, but not all (around 15%), harbouring punctuate accumulation of JMJD6-GFP in the nucleolus in agreement with a rapid and transient recruitment of JMJD6 at rDNA breaks (Fig S2).”

These data are inconclusive as they are. The presented panel in figure S2 should include enlargements of all channels presented above, not only GFP. Furthermore, the 15% of cells with JMJD6 in nucleoli are likely to be caused by the transient expression of JMJD6.Therefore a proper control should be included, either a non-targeting gRNA and/or an catalytic dead version of Cas9 and the number of cells with JMJD6 positive nucleoli should then be quantified and compared.

"Considering the size of the rDNA loci (representing approximately 0.5% of the genomic DNA, from the calculation made by considering that the human genome is composed of three billions base pairs, that one rDNA repeat is approximately 40 kb in size and that there is around 300 rDNA repeats [2]), the irradiation dose we used (a 1 Gy dose generates approximately 20-40 DSB [24]) in the clonogenic cell survival assay should not lead to breaks in the rDNA in a significant proportion of cells."

Is this simple calculation valid? It appears very controversial and to simplistic. Nucleoli occupies a large area of the human nucleus and number of basepairs is likely not a determinant for the area a genomic domain occupies and therefore nor the likelihood of IR to induce a DSB. The authors should remove this argument or provide direct evidence.

"Taken together, this data indicates that the full expression of JMJD6 is specifically required for the management of DSBs occurring at the rDNA locus, consistent with our finding that JMJD6 is always recruited to the nucleolus upon DNA damage."

The last sentence should be toned down as it does not reflect the included data accurately.

"More importantly, these results were confirmed by using CRISPR-Cas9 system targeting DSB at rDNA (Fig 8D and E), indicating that it is not due to a signalisation from DNA breaks induced outside the nucleolus. JMJD6 depletion by siRNA also affected nucleolar caps produced upon induction of breaks in the rDNA in the immortalized MRC5 cell line, indicating that it is not restricted to U2OS cells. (Fig 8E)."

The authors should include representative images showing the decrease in nucleolar caps in U2OS and MRC5 along with the graphs currently included.

"Using the JMJD6-KO cell lines and the WT or Mutant -complemented cell lines, we observed a higher level of interaction between NBS1 and Treacle in JMJD6 KO and Mutant-complemented cell lines compared to WT-complemented cell line after IR exposure (Fig 6 and Fig S7)."

And then….

"As expected, we observed an increase in NBS1 foci in the nucleolus of control cells in response to IR exposure. In contrast, IR-induced nucleolar localization of NBS1 was strongly decreased in JMJD6 KO cells, an effect which was reversed when these cells were complemented with JMJD6 but not in the cells complemented with the inactive form of JMJD6 (Fig 10B)………… These results show that JMJD6 and its enzymatic activity are required for the recruitment of NBS1 to the nucleolus in response to DNA damage, providing a direct link between JMJD6 and the presence of DNA repair factors in the nucleolus."

The data presented is contradictory and the authors need to provide an explanation for the results and what the differences between the results are.

Minor comment:

rDNA; ribosomal RNA genes

Reviewer #3: While the authors have largely addressed most of my concerns, they were not able to address all concerns within the time allowed for resubmission. It is appreciated that the authors attempted to confirm the effects of JMJD6 on the Treacle-NBS1 integration by co-IP, a technique that is often troublesome. While this is still a weakness of the study, the revised study is much improved and has sufficient novelty to be of interest to those interested in the nucleolar DDR. Over all, my concerns have be adequately addressed.

**Have all data underlying the figures and results presented in the manuscript been provided?**

Reviewer #1: Yes

Reviewer #2: Yes

Reviewer #3: Yes

PLOS authors have the option to publish the peer review history of their article (what does this mean?). If published, this will include your full peer review and any attached files.

Reviewer #1: No

Reviewer #2: No

Reviewer #3: No

---

## [Editor Report · Decision Letter 2]

5 May 2020

Dear Dr canitrot,

We are pleased to inform you that your manuscript entitled "JMJD6 participates in the maintenance of ribosomal DNA integrity in response to DNA damage" has been editorially accepted for publication in PLOS Genetics. Congratulations!

Yours sincerely,

Nancy Maizels, Ph.D.

Associate Editor

PLOS Genetics

Gregory Barsh

Editor-in-Chief

PLOS Genetics

Comments from the reviewers (if applicable):

**Data Deposition**

http://datadryad.org/submit?journalID=pgenetics&manu=PGENETICS-D-19-01818R2

**Press Queries**

---

## [Editor Report · Acceptance letter]

19 Jun 2020

PGENETICS-D-19-01818R2 

JMJD6 participates in the maintenance of ribosomal DNA integrity in response to DNA damage 

Dear Dr canitrot, 

We are pleased to inform you that your manuscript entitled "JMJD6 participates in the maintenance of ribosomal DNA integrity in response to DNA damage" has been formally accepted for publication in PLOS Genetics! Your manuscript is now with our production department and you will be notified of the publication date in due course.

With kind regards,

Matt Lyles

PLOS Genetics

On behalf of:
